# An atlas of neural crest lineages along the posterior developing zebrafish at single-cell resolution

**Aubrey GA Howard IV†, Phillip A Baker†, Rodrigo Ibarra-García-Padilla, Joshua A Moore, Lucia J Rivas, James J Tallman, Eileen W Singleton, Jessa L Westheimer, Julia A Corteguera, Rosa A Uribe***

Department of BioSciences, Rice University, Houston, United States

**Abstract** Neural crest cells (NCCs) are vertebrate stem cells that give rise to various cell types throughout the developing body in early life. Here, we utilized single-cell transcriptomic analyses to delineate NCC-derivatives along the posterior developing vertebrate, zebrafish, during the late embryonic to early larval stage, a period when NCCs are actively differentiating into distinct cellular lineages. We identified several major NCC/NCC-derived cell-types including mesenchyme, neural crest, neural, neuronal, glial, and pigment, from which we resolved over three dozen cellular subtypes. We dissected gene expression signatures of pigment progenitors delineating into chromatophore lineages, mesenchyme cells, and enteric NCCs transforming into enteric neurons. Global analysis of NCC derivatives revealed they were demarcated by combinatorial *hox* gene codes, with distinct profiles within neuronal cells. From these analyses, we present a comprehensive cell-type atlas that can be utilized as a valuable resource for further mechanistic and evolutionary investigations of NCC differentiation.

**\*For correspondence:**
rosa.uribe@rice.edu

†These authors contributed equally to this work

**Competing interests:** The authors declare that no competing interests exist.

## Introduction

Unique to vertebrates, neural crest cells (NCC) are an embryonic stem cell population characterized as transient, highly migratory, and multipotent. Following their birth from the dorsal neural tube, NCCs migrate extensively, dorsolaterally or ventrally along the main axial levels of the embryo; the cranial, vagal, trunk, and sacral regions (*Graham et al., 2004*; *Le Douarin and Teillet, 1974*). Depending on the axial level of their origination, NCCs give rise to different cell types within many critical tissues, such as the cornea, craniofacial cartilage and bone, mesenchyme, pigment cells in the skin, as well as neurons and glia that comprise peripheral ganglia (*Hutchins et al., 2018*; *Epstein et al., 1994*; *Kuo and Erickson, 2011*; *Hall and Hörstadius, 1988*; *Le Douarin and Kalcheim, 1999*; *Theveneau and Mayor, 2012*; *Williams and Bohnsack, 2015*; *Yntema and Hammond, 1954*).

During their development, NCCs undergo dramatic transcriptional changes which lead to diverse cellular lineages, making their transcriptomic profiles highly dynamic (*Simoes-Costa et al., 2014*; *Martik and Bronner, 2017*; *Soldatov et al., 2019*; *Williams et al., 2019*). In support of the model that complex transcriptional programs govern NCC ontogenesis, gene regulatory networks involved in early development of NCCs into broad cell types have been studied at a high level using a combination of transcriptomics, chromatin profiling, and enhancer studies, especially during pre-migratory and early migratory NCC specification along cranial axial regions, across amniotes (*Martik and Bronner, 2017*; *Simoes-Costa and Bronner, 2016*; *Green et al., 2015*; *Lumb et al., 2017*; *Williams et al., 2019*; *Hockman et al., 2019*). For example, during pre-migratory stages the transcription factors FoxD3, Tfap2a, and Sox9 are important for NCC fate specification and in turn regulate the expression of Sox10, a conserved transcription factor that is expressed along all axial levels

by early migrating NCCs and within many differentiating lineages (*Sauka-Spengler and Bronner-Fraser, 2008*; *Martik and Bronner, 2017*). Gene regulatory networks that are important for select NCC cell fates, like melanocytes and chondrocytes, have been well characterized (reviewed in *Martik and Bronner, 2017*). Recently, the regulatory circuitry behind glial, neuronal, and mesenchymal fates of vagal NCC was described (*Ling and Sauka-Spengler, 2019*) where *Prrx1* and *Twist1* have been described as key differentiation genes for mesenchymal fate. Despite this progress, however, comprehensive knowledge of the genes that are expressed and participate in NCC lineage differentiation programs during later phases of embryogenesis remains to be fully characterized, particularly for posterior tissues (reviewed in *Hutchins et al., 2018*). Indeed, altered gene expression during NCC differentiation can cause several neurocristopathies, such as DiGeorge syndrome, neuroblastoma, Hirschsprung disease, Auriculo-condylar syndrome, and Klein-Waardenburg syndrome (*Barlow, 1984*; *Bolande, 1997*; *Brosens et al., 2016*; *Escot et al., 2016*; *Vega-Lopez et al., 2017*; *Wang et al., 2014*), further highlighting the need to understand NCC spatiotemporal gene expression patterns during their differentiation into diverse cellular types.

Previous single-cell transcriptomic studies in zebrafish have laid a strong foundation to globally map early lineages of a majority of cell types through early to middle embryonic development (*Wagner et al., 2018*; *Tambalo et al., 2020*), and recently this has been extended into the larval stage (*Farnsworth et al., 2020*). With respect to zebrafish NCC development, the early embryonic window of 11–20 hr post fertilization (hpf) marks the stage of NCC specification and the emergence of their migratory behavior. Further development between 24 and 96 hpf represents the time when NCCs actively differentiate into their many derivatives (*Rocha et al., 2020*). Concerning the posterior NCC fates, however, many of these cells undergo differentiation programs during the embryonic to larval transition, a developmental stage that emerges between ~48 and 72 hpf. Transcriptomic analysis during this transitional phase would therefore enhance our understanding of the dynamic shifts in cell states that may regulate cellular differentiation programs.

In this study, we leverage the power of single-cell transcriptomics and curate the cellular identities of *sox10*-expressing and *sox10*-derived populations along the posterior zebrafish during development. We have utilized the Tg($-4.9sox10$:EGFP, hereafter referred to as *sox10*:GFP) transgenic line to identify NCCs and their recent derivatives (*Carney et al., 2006*). Using *sox10*:GFP$^+$48–50 hpf embryos and 68–70 hpf larvae, we identified eight major classes of cells: mesenchyme, NCC, neural, neuronal, glial, pigment, muscle, and otic. Among the major cell types, we annotated over 40 cellular subtypes. By leveraging in depth analysis of each time point separately, we captured the dynamic transition of several NCC fates, most notably we discovered over a dozen transcriptionally distinct mesenchymal subpopulations and captured the progressive differentiation of enteric neural progenitors into maturing enteric neurons. Using Hybridization Chain Reaction (HCR) and in situ hybridization, we validated the spatiotemporal expression patterns of various subtypes. By merging our 48–50 hpf and 68–70 hpf datasets, we generated a comprehensive atlas of *sox10*$^+$ cell types spanning the embryonic to larval transition, which can also be used as a tool to identify novel genes and mechanistically test their roles in the developmental progression of posterior NCCs. Using the atlas, we characterized a *hox* signature for each cell type, detecting novel combinatorial expression of *hox* genes within specific cell types. Our intention is that this careful analysis of posterior NCC fates and resulting atlas will aid the cell and developmental biology communities by advancing our fundamental understanding of the diverging transcriptional landscape during the NCC's extensive cell fate acquisition.

## Results

### Single-cell profiling of sox10:GFP$^+$ cells along the posterior zebrafish during the embryonic and larval stage transition

To identify *sox10*-expressing and *sox10*-derived cells along the posterior zebrafish during the embryonic to larval transition, we utilized the transgenic line *sox10*:GFP (*Figure 1*; *Carney et al., 2006*; *Kwak et al., 2013*). Tissue posterior to the otic vesicle, encompassing the vagal and trunk axial region (*Figure 1B*), was dissected from 100 embryonic zebrafish at 48–50 hpf and 100 larval zebrafishes at 68–70 hpf. Dissected tissues were dissociated and immediately subjected to fluorescence-activated cell sorting (FACS) to isolate *sox10*:GFP$^+$ cells (*Figure 1B*; *Figure 1—figure supplement*

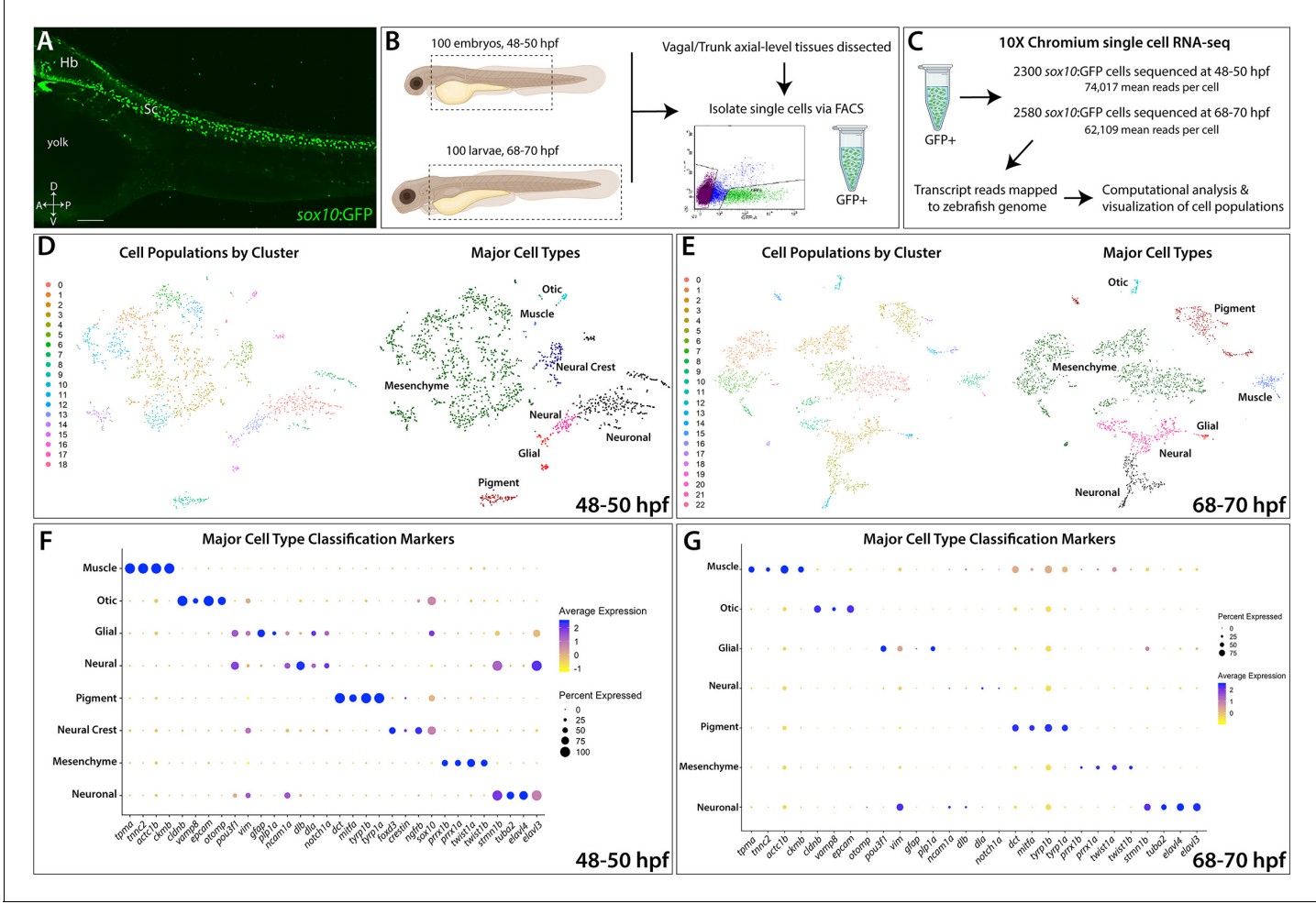

**Figure 1.** Single-cell profiling strategy and cell population composition of posterior *sox10*:GFP[+] cells from the posterior zebrafish during the embryonic to larval stage transition. (**A**) Confocal image of a *sox10*:GFP[+] embryo at 48 hpf; Hb: Hindbrain; Sc: Spinal cord. A: Anterior, P: Posterior, D: Dorsal, V: Ventral. Scale bar: 50 μM (**B**) Cartoon illustrations of a zebrafish embryo at 48–50 hpf and an early larval fish at 68–70 hpf depicted laterally to summarize the dissection workflow used to collect posterior *sox10*:GFP[+] cells. (**C**) Schematic of the 10X Genomics Chromium and data analysis pipeline. (**D**) tSNE plots showing the arrangement of Clusters 0–18 and where the major cell types identified among *sox10*:GFP[+] cells arrange in the 48–50 hpf dataset. (**E**) tSNE plots showing the arrangement of Clusters 0–22 and where the major cell types identified among *sox10*:GFP[+] cells arrange in the 68–70 hpf dataset. (**F,G**) Dot plots of the identifying gene markers for each major cell type classification in the 48–50 hpf and 68–70 hpf datasets, respectively. Dot size depicts the cell percentage for each marker within the dataset and the color summarizes the average expression levels for each gene.

The online version of this article includes the following source data and figure supplement(s) for figure 1:

**Source data 1.** List of marker genes per cluster in the *sox10*:GFP scRNA-seq datasets.

**Source data 2.** Table summarizing the top identity markers used for major cell type and subtype cellular classifications for each cluster at 48–50 and 68–70 hpf.

**Figure supplement 1.** Statistics on generation and filtering of single cell transcriptomes at 48–50 hpf and 68–70 hpf.

**Figure supplement 2.** Major cell type annotations among *sox10*:GFP[+] cells.

**Figure supplement 3.** Major cell type categories and cell cycle distributions of the scRNA-seq datasets.

**Figure supplement 4.** Identification of otic vesicle, muscle, and central nervous system (CNS) cellular populations.

**Figure supplement 5.** Identification of fin bud and sensory neuronal progenitor cellular populations.

*1A,B*). Isolated cells were then input into 10X Genomics Chromium scRNA-seq assays and captured at a depth of 2300 cells from the 48–50 hpf time point and 2580 cells from the 68–70 hpf time point (*Figure 1C*; *Figure 1—figure supplement 1C*). We performed cell filtering and clustering (*Figure 1—figure supplement 1D–I*) of the scRNA-seq datasets using Seurat (*Butler et al., 2018*; *Stuart et al., 2019*) to computationally identify cell populations based on shared transcriptomes, yielding 1608

cells from the 48–50 hpf time point and 2410 cells from the 68–70 hpf time point, totaling 4018 cells for final analysis (*Figure 1—figure supplement 1C*). We detected cell population clusters with transcriptionally unique signatures, as shown in heatmap summaries that revealed the top enriched gene signatures per cluster, with 19 clusters (0–18) from the 48–50 hpf time point (*Figure 1—figure supplement 2A*) and 23 clusters (0–22) from the 68–70 hpf time point (*Figure 1—figure supplement 2B*), totaling 42 clusters across both time points. Datasets were visualized with the t-Distributed Stochastic Neighbor Embedding (tSNE) method, which spatially grouped cells in each cluster, for both time points examined (*Figure 1D,E*). The top significantly enriched markers for each cluster at 48–50 and 68–70 hpf are provided in a table in *Figure 1—source data 1*.

## Major classification of sox10:GFP⁺ cell states

To assess the proliferative state of *sox10*:GFP⁺ cells, we determined their G1, S or G2/M phase occupancy, based on expression of proliferative cell cycle marker genes (*Figure 1—figure supplement 3I*). At 48–50 hpf, 52% of *sox10*:GFP⁺ cells were in G1 phase, 31% were in the S phase and 17% in G2/M phase (*Figure 1—figure supplement 3G*), collectively indicating that 48% of the cells in the 48–50 hpf time point were proliferative. At 68–70 hpf, 64% of cells were in G1 phase, 24% of cells were in the S phase and 12% in G2/M phase (*Figure 1—figure supplement 3G*), indicating that 36% of the cells were proliferative. The cell cycle occupancy distributions were visualized in tSNE plots, revealing congregations of proliferative and non-proliferative *sox10*:GFP⁺ cells (*Figure 1—figure supplement 3A,B*); *aurkb* and *mcm3* confirmed general occupancy in the G2/M and S phase (*Figure 1—figure supplement 3C–F*). Together, these data of cell cycle state reflect a general decrease in proliferative cells among *sox10*:GFP⁺ populations between 48 and 70 hpf, in agreement with prior observations (*Rajan et al., 2018*).

Using a combination of gene expression searches of literature and bioinformatics sources, examination of the scRNA-seq transcriptomes indicated that *sox10*:GFP⁺ cells exist in several major cell type categories based on the expression of signature marker genes (*Figure 1F,G*; *Figure 1—figure supplement 2E,F*). These major cell type categories included: neural, neuronal, glial, mesenchyme, pigment cell, NCC, otic, and muscle; their respective fraction of the datasets was also calculated (*Figure 1D–G*; *Figure 1—figure supplement 2C,D*; *Figure 1—figure supplement 3H*). Neuronal identity refers to cells predominantly expressing neuron markers, such as *elavl3/4*, while neural cells are defined by a multipotent state with potential towards fates of either glial or neuron identity, and marked by expression of factors such as *sox10*, *dla*, and/or *ncam1a*. Notably, mesenchyme identity represented the largest proportion of the datasets at 61% and 53% of the cells at 48–50 and 68–70 hpf, respectively (*Figure 1—figure supplement 3H*). Mesenchyme clusters were identified by a combination of mesenchymal gene markers including *twist1a/b* and *prrx1a/b* (*Soldatov et al., 2019*). In addition, cells with an otic vesicle and muscle identity were detected (*Figure 1D–G*; *Figure 1—figure supplement 3H*; *Figure 1—figure supplement 4*), as has previously been described in the *sox10*:GFP line (*Carney et al., 2006*; *Rajan et al., 2018*; *Rodrigues et al., 2012*; *Kwak et al., 2013*). Overall, major cell type cluster identities and their top signature marker genes are summarized in *Figure 1—source data 2*.

## Annotation of cellular types among posterior sox10:GFP⁺ cells

Closer analysis of the 42 cluster gene signatures among the two time points allowed us to annotate cellular identities in more detail (*Figure 1—source data 2*). Indeed, we identified previously described NCC-derived cell types. For example, the *sox10*:GFP line has been shown to transiently label sensory dorsal root ganglion (DRG) progenitors between the first and second day of zebrafish development (*McGraw et al., 2008*; *Rajan et al., 2018*). We observed sensory neuronal/DRG gene expression in Cluster 17 at 48–50 hpf (*Figure 1—source data 2*; *Figure 1—figure supplement 5*) by the markers *neurod1*, *neurod4*, *neurog1*, *six1a/b*, *elavl4* (*Carney et al., 2006*; *Delfino-Machín et al., 2017*). Additionally, we identified other NCC-derivatives, including mesenchymal cells (*Le Lièvre and Le Douarin, 1975*; *Kague et al., 2012*; *Soldatov et al., 2019*; *Ling and Sauka-Spengler, 2019*), pigment cells (*Reedy et al., 1998*; *Higdon et al., 2013*), and enteric neurons (*Kelsh and Eisen, 2000*; *Kuo and Erickson, 2011*; *Lasrado et al., 2017*), which we describe in further detail for both time points in *Figures 2–5*.

# Identification of pigment cell types within sox10:GFP⁺ scRNA-seq datasets

With robust genetic lineage details published on pigment cell differentiation in zebrafish (*Kelsh, 2004*; *Lister, 2002*; *Quigley and Parichy, 2002*), we sought to validate the scRNA-seq datasets by assessing if we could resolve distinct pigment cell populations. Pigment cell development has been broadly studied in the developing zebrafish, where NCCs give rise to three distinct chromatophore populations: melanophores, xanthophores, and iridophores (*Figure 2A*). Our annotation analysis of *sox10*:GFP⁺ scRNA-seq clusters revealed expression of pigment cell lineage gene markers (*Figure 2B–J*; *Figure 1—figure supplement 2*). At 48–50 hpf, melanophores were detected in Cluster eight based on expression of *mitfa*, *dct*, *tyrp1b*, and *pmela* (*Du et al., 2003*; *Lister et al., 1999*; *Ludwig et al., 2004*; *Quigley and Parichy, 2002*; *Figure 1F*; *Figure 2A*), also reflected by the dot

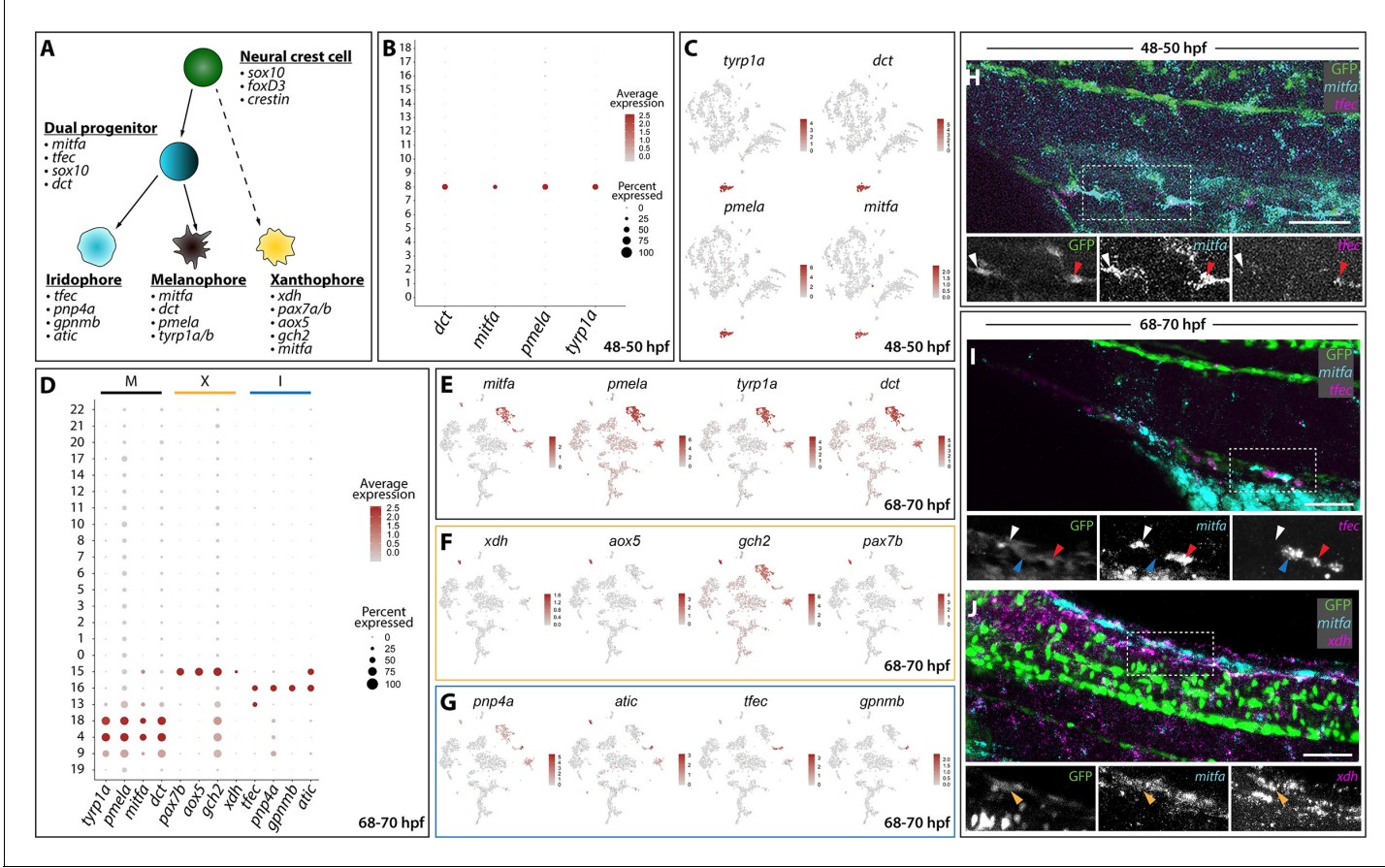

**Figure 2.** Distinct pigment cell populations are present among *sox10*:GFP⁺ cells during embryonic to larval transition. (A) Cartoon schematic depicting the model for neural crest delineation into pigment cell lineages and the genes that were used to identify each pigment cell population. (B) Dot plot identifying melanophore markers within the 48–50 hpf dataset. Dot size depicts the cell percentage for each marker within the dataset and the color summarizes the average expression levels for each gene. (C) tSNE plots depicting melanophore signature in the 48–50 hpf dataset. Relative expression levels are summarized within the color keys, where color intensity is proportional to expression level of each gene depicted. (D) Dot plot showing distinct pigment chromatophore markers within the 68–70 hpf dataset. Dot size depicts the cell percentage for each marker within the dataset and the color summarizes the average expression levels for each gene. M: melanophore markers; X: xanthophore markers; I: iridophore markers. (E–G) tSNE plots revealing the location of melanophores (E), xanthophores (F), and iridophores (G) in the 68–70 hpf dataset. Relative expression levels are summarized within the color keys, where color intensity is proportional to expression level of each gene depicted. (H) HCR against *mitfa* and *tfec* at 48–50 hpf reveals *mitfa*⁺ melanophores (white arrowhead) and *mitfa*⁺/*tfec*⁺ pigment progenitors (red arrowhead). Cropped panels show individual fluorescent channels. (I) HCR against *mitfa* and *tfec* at 68–70 hpf presents *mitfa*⁺ melanophores (white arrowhead), *tfec*⁺ iridophores (blue arrowhead), and *mitfa*⁺/*tfec*⁺ pigment progenitors (red arrowhead). Cropped panels show individual fluorescent channels. (J) HCR against *mitfa* and *xdh* at 68–70 hpf shows *mitfa*⁺/*xdh*⁺ xanthophores (orange arrowhead). Cropped panels show individual fluorescent channels. Scale bar in H-J: 50 μm.

The online version of this article includes the following source data for figure 2:

**Source data 1.** Melanophore populations, shared and unique genes at 68–70 hpf.

and tSNE plots (*Figure 2B,C*). At 68–70 hpf, we resolved discrete pigment cell populations that included xanthophore, iridophore, and two distinct melanophore clusters (*Figure 2D–G*, *Figure 1—source data 2*). The xanthophores mapped to Cluster 15 and were enriched with *xdh*, *aox5*, *pax7b*, *mitfa*, and *gch2* (*Nord et al., 2016*; *Parichy et al., 2000*; *Saunders et al., 2019*; *Minchin and Hughes, 2008*; *Lister et al., 1999*; *Figure 2A,D,F*). Cluster 16 was identified as iridophores, which presented the well-characterized markers: *tfec*, *pnp4a*, *gpnmb*, and *atic* (*Higdon et al., 2013*; *Lister et al., 2011*; *Petratou et al., 2018*; *Petratou et al., 2019*; *Figure 2A,D,G*; *Figure 1—source data 2*). The use of cell cycle markers revealed that two different melanophore clusters at 68–70 hpf (Clusters 4 and 18) were present in different proliferative states (*Figure 1—source data 2*). While the majority of cells in Cluster four were in G1, Cluster 18 expressed S and G2/M markers, such as *pcna* and *aurkb*, suggesting this population to be proliferating melanophores (*Figure 1—figure supplement 3B,E,F*; *Figure 2—source data 1*).

At 68–70 hpf, we identified a pigment progenitor population, where iridophore and melanophore markers were co-expressed in Cluster 13 (*Figure 2A,D*). These undifferentiated pigment progenitor cells expressed *tfec* in combination with *mitfa* and have been described recently at 24, 30, and 48 hpf (*Petratou et al., 2018*). Additionally, Cluster 13 expressed *tfap2e*, *gpx3*, and *trpm1b* (*Figure 1—source data 2*) whose expression patterns have been previously reported in pigment progenitors (*Saunders et al., 2019*). Finally, a population of pigmented muscle (Cluster 9) was also found with a weak melanophore signature coupled with expression of the muscle markers *ckmb*, *tpma*, *tnnc2*, and *tnnt3b* (*Figure 1—source data 2*; *Figure 1—figure supplement 4*).

We next performed whole mount HCR to assess the spatial co-expression of *mitfa*, *tfec* and *xdh*. When examining *mitfa* and *tfec* at 48–50 hpf (*Figure 2H*), we detected *sox10*:GFP[+] cells that expressed *mitfa*, identifying the melanophores (*Figure 2H*; white arrowhead), and cells that expressed both *mitfa* and *tfec*, defining the pigment progenitors (*Figure 2H*; red arrowhead). At 68–70 hpf, we confirmed the four distinct pigment populations we identified through Seurat (*Figure 2B–G*):GFP[+] melanophores expressing *mitfa* only (*Figure 2I*; white arrowhead), iridophores only expressing *tfec* (*Figure 2I*; blue arrowhead), and pigment progenitors expressing both *mitfa* and *tfec* (*Figure 2I*; red arrowhead) were detected. When examining *xdh* and *mitfa* expression patterns, *sox10*:GFP[+] xanthophores were found to be expressing both markers (*Figure 2J*; orange arrowhead).

Taken together, the above-described results regarding pigment cell expression patterns validates that the *sox10*:GFP[+]scRNA-seq datasets captured discrete NCC-derived populations, and coupled with HCR analysis, shows we are able to validate these cell populations in vivo.

## Mesenchyme in the posterior embryo and larvae exists in various transcriptionally-distinct populations

Heatmap analysis of gene expression groups depicted that mesenchyme cells clustered together globally within the datasets (*Figure 1—figure supplement 2C,D*; *Figure 3A,B*), with *twist1a* expression broadly labeling all mesenchyme cells (*Figure 1—figure supplement 2E,F*). In addition to *twist1a*, mesenchyme cells also expressed *prrx1a/b*, *twist1b*, *foxc1a/b*, *snai1a/b*, *cdh11*, *sparc*, *colec12*, *meox1*, *pdgfra* (*Figure 3A,B*), and other known mesenchymal markers such as *mmp2* (*Figure 1—source data 2*; *Janssens et al., 2013*; *Theodore et al., 2017*). In whole mount embryos at 48 hpf, we observed broad expression of *foxc1a* and *mmp2* along the posterior pharyngeal arches and ventral regions of the embryo via in situ hybridization (*Figure 3—figure supplement 1C,D*; arrowheads), confirming their expression territories within posterior-ventral mesenchymal tissues.

Further analysis revealed various transcriptionally-distinct populations were present among the *sox10*:GFP[+] cells with a mesenchymal identity. Among these, we detected nine clusters with chondrogenic signatures—identified by expression of mesenchymal signature genes, as well as the chondrogenic markers *barx1* and/or *dlx2a* (*Sperber et al., 2008*; *Sperber and Dawid, 2008*; *Ding et al., 2013*; *Barske et al., 2016*; *Figure 3C,D*). Feature plot exports revealed distribution of the chondrogenic cells (*barx1*[+]) in relation to all other mesenchyme (*prrx1b*[+], *twist1a*[+]) cells in the datasets (*Figure 3E,L*). Within the nine chondrogenic clusters, we discovered gene expression indicative of heterogeneous cell states, ranging from proliferative, progenitor/stem-like, and migratory to differentiating signatures (*Figure 1—source data 2*). All other mesenchyme clusters (seven in total) were also classified into various progenitor and differentiation categories. Among these categories, the clusters expressed either proliferative progenitor markers, differentiation signatures, or general

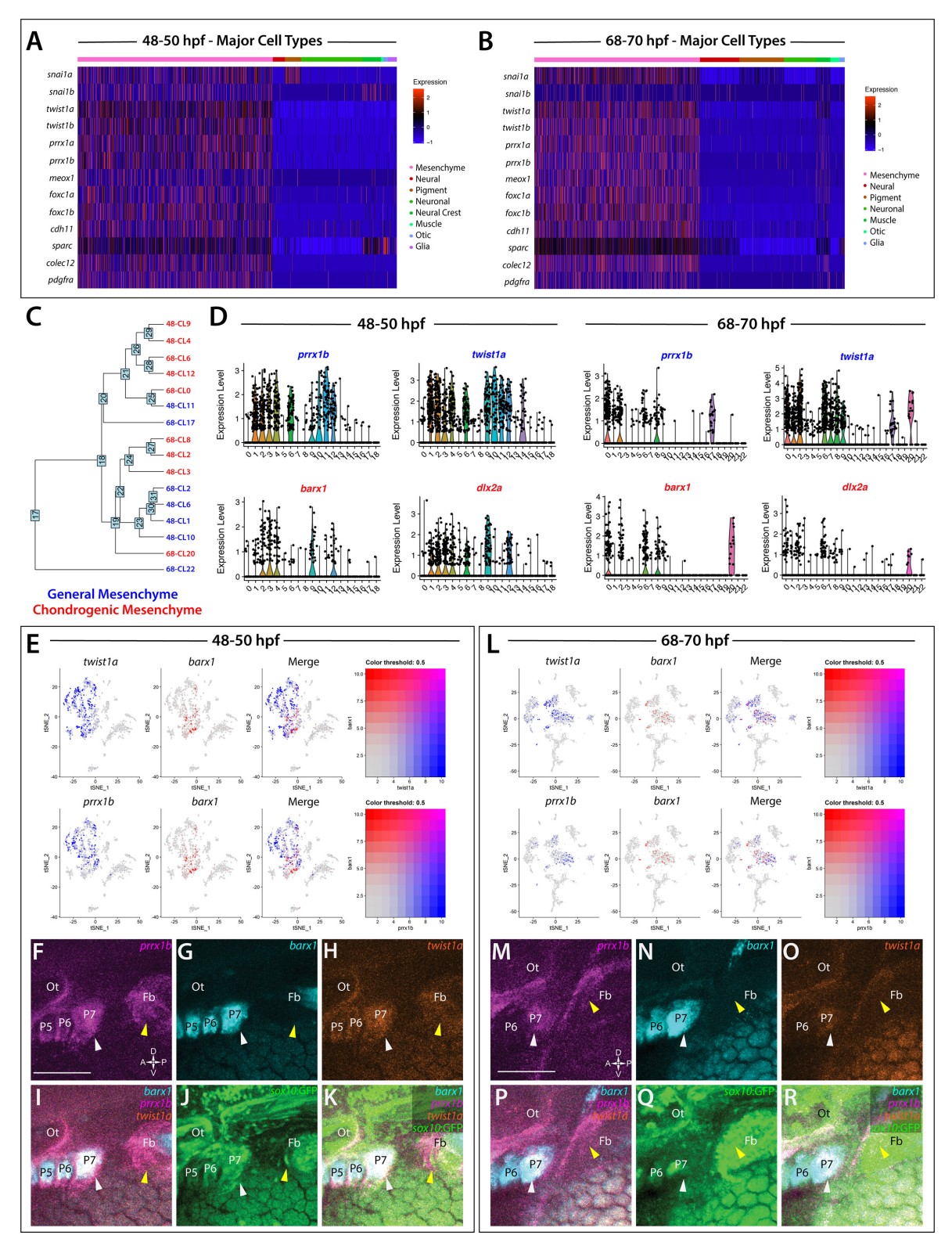

**Figure 3.** Global analysis of mesenchyme cell signatures among *sox10*:GFP⁺ cells. (A,B) A heatmap of signature mesenchyme identity genes within the major cell type classified cells at 48–50 and 68–70 hpf, respectively. Relative expression levels within each cluster is summarized within the color key, where red to blue color indicates high to low gene expression levels. (C) A cluster tree depicting the relationship between general and chondrogenic mesenchyme cellular subtypes. (D) Violin plots summarizing the expression levels for select mesenchyme identity markers within individual clusters at

*Figure 3 continued on next page*

*Figure 3 continued*

the 48–50 and 68–70 hpf time points, respectively. Data points depicted in each cluster represent single cells expressing each gene shown. (**E,L**) tSNE plots depicting the co-expression of *twist1a* (blue) and *barx1* (red) or *prrx1b* (blue) and *barx1* (red) in the 48–50 and 68–70 hpf datasets, respectively. Relative expression levels are summarized within the color keys, where color intensity is proportional to expression level of each gene depicted. (**F–K**) Whole mount HCR analysis reveals the spatiotemporal expression of *prrx1b* (**F**), *barx1* (**G**), *twist1a* (**H**), *sox10*:GFP (**J**) in 48 hpf embryos. (**I**) A merge of *barx1*, *prrx1b*, and *twist1a* is shown. (**K**) A merge of *barx1*, *prrx1b*, *twist1a*, and *sox10*:GFP is shown. White arrowheads denote expression in posterior pharyngeal arch, while yellow arrowheads highlight fin bud expression. (**M–R**) Whole mount HCR analysis reveals the spatiotemporal expression of *prrx1b* (**M**), *barx1* (**N**), *twist1a* (**O**), *sox10*:GFP (**Q**) in 68 hpf embryos. (**P**) A merge of *barx1*, *prrx1b*, and *twist1a* is shown. (**R**) A merge of *barx1*, *prrx1b*, *twist1a*, and *sox10*:GFP is shown. White arrowheads denote expression in posterior pharyngeal arch, while yellow arrowheads highlight fin bud expression. Ot: otic; Fb: Fin bud. Scale bar: 100 μm.

The online version of this article includes the following figure supplement(s) for figure 3:

**Figure supplement 1.** Whole mount in situ hybridization of select ENCC, mesenchyme and neural markers at 48–50 hpf.

migratory mesenchymal markers (*Figure 1—source data 2*). Cluster 14 at 48 hpf and Clusters 1 and 7 at 68–70 hpf exhibited a general mesenchymal signature, but also expressed fin bud marker genes (*hand2, tbx5a, hoxd13a, prrx1a*, *Figure 1—figure supplement 5*; *Yelon et al., 2000*; *Lu et al., 2019*; *Nakamura et al., 2016*; *Feregrino et al., 2019*). Additionally, visualization of clusters with general mesenchyme and chondrogenic identities using a cluster tree highlighted potential similar clusters between the time points (*Figure 3C*). For example, the cluster tree showed proximal location of Cluster 8 at 68–70 hpf and Cluster 2 at 48–50 hpf, which we noted contained clear proliferative chondrogenic gene signatures (*Figure 3D*; *Figure 1—source data 2*).

To confirm the spatial co-expression of *prrx1b*, *twist1a*, and *barx1* within *sox10*:GFP⁺ tissues, we utilized HCR analysis (*Figure 3F–K,M–R*). Corroborating our analysis that mesenchyme-identity populations contained both general and chondrogenic signatures, we found the co-expression of *prrx1b*, *twist1a*, and *barx1* within *sox10*:GFP⁺ domains along the posterior pharyngeal arches (white arrowheads) and fin bud mesenchyme (yellow arrowheads) at both time points (*Figure 3F–K,M–R*).

Overall, the above-described analyses indicate that *sox10*:GFP⁺ mesenchymal cells in the posterior zebrafish exhibit various transcriptional states between the embryonic to larval transition and suggest that posterior mesenchyme exists in various subpopulations during its differentiation.

## sox10-derived cells during the embryonic to early larval transition reveal enteric progenitor to enteric neuron progression

At 48–50 hpf, cells with NCC identity were notably detected in Cluster 5, defined by expression of the core NCC markers *sox10, foxd3, crestin*, and *tfap2a* (*Figure 1F*; *Figure 4*, *Figure 1—source data 2*; *Dutton et al., 2001*; *Luo et al., 2001*; *Knight et al., 2003*; *Stewart et al., 2006*). Moreover, Cluster five was found to contain various other genes previously shown to be expressed in zebrafish NCCs; including, *vim, snai1b, sox9b, zeb2a, mych*, and *mmp17b* (*Figure 4D*; *Cerdà et al., 1998*; *Heffer et al., 2017*; *Hong et al., 2008*; *Leigh et al., 2013*; *Van Otterloo et al., 2012*; *Wang et al., 2011*; *Rocha et al., 2020*). We reasoned that many of the NCCs had started their respective differentiation programs and were beginning to assume specified lineage profiles. Therefore, we sought to determine if the NCC cluster also contained gene expression profiles of known differentiating NCC types along the posterior body, such as enteric progenitors, also known as enteric neural crest cells (ENCCs).

ENCCs fated to give rise to the enteric nervous system (ENS), the intrinsic nervous system within the gut, express a combination of NCC and enteric progenitor marker genes over developmental time (reviewed in *Nagy and Goldstein, 2017*; *Rao and Gershon, 2018*), which occurs between 32 and 72 hpf in zebrafish (reviewed in *Ganz, 2018*). Enteric markers in zebrafish include *sox10, phox2bb, ret, gfra1a, meis3*, and *zeb2a* (*Dutton et al., 2001*; *Shepherd et al., 2004*; *Elworthy et al., 2005*; *Delalande et al., 2008*; *Heanue and Pachnis, 2008*; *Uribe and Bronner, 2015*). Therefore, we expected to capture a population of ENCCs within our 48–50 hpf dataset. Indeed, within Cluster 5 we observed expression of the enteric markers *phox2bb, ret, gfra1a, meis3, sox10*, and *zeb2a* (*Figure 4B–D*). Using whole mount in situ hybridization, we confirmed the expression of *sox10* and *phox2bb* within ENCCs localized along the foregut at 48 hpf (*Figure 3—figure supplement 1A,A',B,B'*; arrowheads). Furthermore, gene orthologs known to be expressed in ENCC in amniotes were detected within Cluster 5, such as *ngfrb* (orthologue to p75;

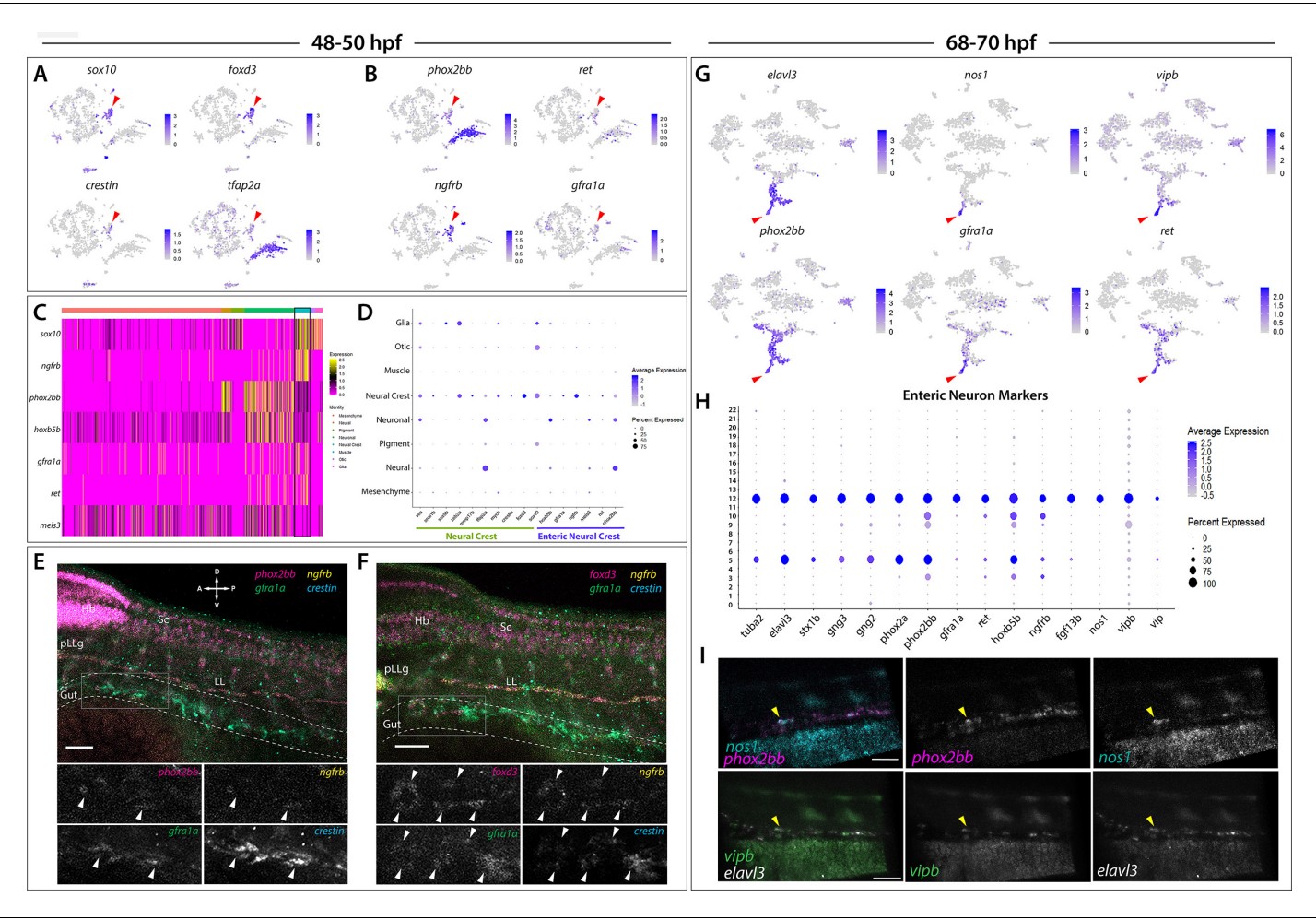

**Figure 4.** Enteric neural crest cells and differentiating enteric neurons are present among posterior *sox10*:GFP⁺ cell populations. (A) tSNE feature plots reveal expression of core neural crest cell markers *sox10*, *foxd3*, *crestin*, and *tfap2a* mapping to the neural crest cell cluster (red arrowhead). (B) tSNE feature plots depict expression of the enteric neural crest cell markers *phox2bb*, *ret*, *ngfrb* and *gfra1a* within the neural crest cell cluster (red arrowhead). Relative expression levels are summarized within the color keys in (A) and (B), where color intensity is proportional to expression level of each gene depicted. (C) A heatmap reveals expression levels of enteric neural crest cell markers across the eight major cell populations captured in the 48–50 hpf data set (color key denotes cells types represented in color bar on top of heatmap). Neural crest cell cluster highlighted in black rectangle. Relative expression levels within each major cell type cluster is summarized within the color key, where yellow to magenta color indicates high to low gene expression levels. (D) Dot plot of expanded list of neural crest (green line) and enteric neural crest (purple line) cell markers across each major cell type within 48–50 hpf data set. Dot size depicts the cell percentage for each marker within the data set and the color summarizes the average expression levels for each gene. (E,F) Whole mount HCR analysis of 48 hpf embryos reveals co-expression of the enteric neural crest cell markers *phox2bb*, *ngfrb*, *gfra1a*, and *crestin* in (E), or *foxd3*, *ngfrb*, *gfra1a* and *crestin* in (F), within the developing gut (dashed outline). Top panels depict merged images of color channels for each HCR probe. Lower panels represent gray-scale images of each separated channel corresponding to the magnified region of foregut (gray rectangle). Arrowheads depict regions where all markers are found to be co-expressed. Hb: Hindbrain, Sc: Spinal cord, pLLg: posterior Lateral Line ganglia, LL: Lateral Line. A: Anterior, P: Posterior, D: Dorsal, V: Ventral. Scale bar: 50 μM. (G) tSNE feature plots reveal expression levels of enteric neuron markers *elavl3*, *phox2bb*, *gfra1a*, *nos1*, *vipb*, and *ret*, within a common region of a neuronal cluster (red arrowhead). Relative expression levels are summarized within the color keys, where color intensity is proportional to expression level of each gene depicted. (H) Dot plot depicts expression levels of pan-neuronal and enteric neuron specific markers across individual clusters generated within the original 68–70 hpf tSNE. Pan-neuronal markers found throughout Clusters 5 and 12, with enteric neuron markers most prominently expressed within Cluster 12. Dot size depicts the cell percentage for each marker within the data set and the color summarizes the average expression levels for each gene. (I) Whole mount HCR analysis depicts differentiating enteric neurons within the foregut region at 69 hpf co-expressing *nos1*, *phox2bb*, *vipb*, and *elavl3* (yellow arrowheads). Anterior: Left, Posterior: Right. Scale bar: 50 μM.

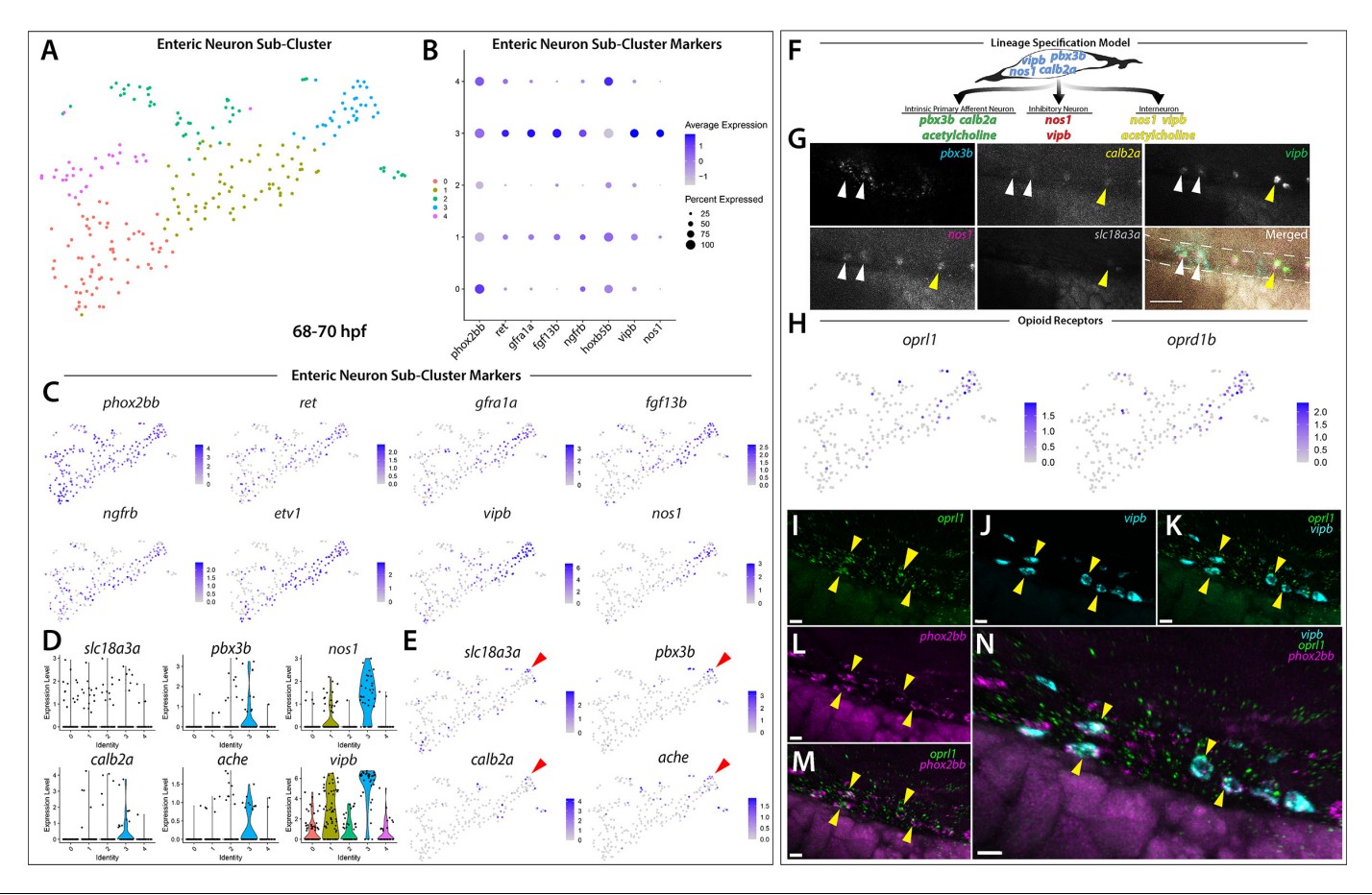

**Figure 5.** Differentiating enteric neurons captured during key transitional stage of subtype diversification within 68–70 hpf *sox10*:GFP⁺ larval cells. (**A**) tSNE plot reveals five distinct sub-clusters following the subset analysis and re-clustering of Clusters 5 and 12 from the 68–70 hpf data set. (**B**) Dot plot depicts expression levels of enteric neuron markers across resulting Sub-clusters. Each marker was expressed at low levels in Sub-cluster 1 and were found to be expressed at higher levels within Sub-cluster 3. (**C**) tSNE feature plots further depict the expression of enteric neuron markers by illustrating the levels and localization of expression within the Sub-cluster architecture. Feature plots supplement dot plot and demonstrate the prominent expression of enteric neuron markers within Sub-cluster 3, which appears to emanate from Sub-cluster 1. (**D,E**) Violin and feature plots reveal expression levels of acetylcholine-associated and excitatory neuron markers reported to distinguish enteric IPANs. These markers were found in a discrete pocket of cells forming the distal-most region of Sub-cluster 3 (red arrowhead). Violin data points depicted in each Sub-cluster represent single cells expressing each gene shown. (**F**) Graphical model summarizes expression patterns observed in 68–70 hpf data set and HCR validation. Common enteric neuroblast capable of diverging into subsequent lineages, IPAN, inhibitory neuron, and interneuron through lineage restricted gene expression. *pbx3b* promotes assumption of IPAN role through loss of *nos1* and *vipb* and begins expressing *calb2a*, *ache*, and *slc18a3a*. (**G**) Whole mount HCR analysis reveals co-expression of IPAN marker genes, *pbx3b* and *calb2a*, and inhibitory neurochemical marker genes, *vipb* and *nos1* (white arrowheads), within the foregut (dashed white line) at 68 hpf. Vesicular acetylcholine transferase, *slc18a3a,* was not observed in tandem with *pbx3b* but was co-expressed with *calb2a, vipb,* and *nos1* (yellow arrowheads). Scale bar: 50 μM. (**H**) Feature plots reveal expression of opioid receptor genes, *oprl1* and *oprd1b*, within the differentiated enteric neuron Sub-cluster 3. (**I–N**) Whole mount HCR analysis validates expression of *oprl1* in combination with *vipb* and *phox2bb* (yellow arrowheads) in enteric neurons localized to the foregut region of a 68 hpf embryo. Scale bar: 10 μM.

The online version of this article includes the following source data and figure supplement(s) for figure 5:

**Source data 1.** List of marker genes per Sub-cluster, following subset and re-clustering of enteric Clusters 5 and 12 at 68–70 hpf.

**Source data 2.** List of enriched pathways within enteric neuron Sub-cluster 3 and genes present in specific opioid proenkephalin pathway identified following PANTHER Overrepresentation Test.

**Figure supplement 1.** Enteric neuron subtype diversification gene expression patterns seen in enteric neuron Sub-clusters related to *Figure 5D,E* Panel of tSNE feature plots magnified and cropped to focus on progressively differentiating enteric neurons (highlighted by *etv1* expression).

**Figure supplement 2.** Enteric and sympathetic neuron markers distinguished among common autonomic neuron precursors.

*Anderson et al., 2006*; *Wilson et al., 2004*) and *hoxb5b* (orthologous to *Hoxb5*; *Kam and Lui, 2015*; *Figure 4B–D*).

HCR analysis of 48 hpf embryos validated the co-expression profiles of several ENCC markers along the foregut (*Figure 4E–F*; foregut in gray box). We observed that a chain of *crestin*[+] cells localized in the foregut contained a subpopulation of cells expressing *ngfrb*, *phox2bb*, and *gfra1a* (*Figure 4E*; white arrowheads), or expressing *foxd3*, *ngfrb*, and *gfra1a* (*Figure 4F*; white arrowheads). Together, these HCR data confirm that ENCC markers are co-expressed along the zebrafish gut.

We next asked if we could resolve discrete differentiating enteric neurons over time. Within the 68–70 hpf zebrafish, ENCCs have yet to finish their migratory journey along the gut and have previously been shown to exist in varying stages of neuronal differentiation, where the earliest differentiating neurons are found in the rostral foregut, and the more proliferative undifferentiated ENCCs are continuing to migrate into the caudal hindgut (*Elworthy et al., 2005*; *Olden et al., 2008*; *Harrison et al., 2014*; *Uribe and Bronner, 2015*; *Taylor et al., 2016*). During early neuronal differentiation (68–70 hpf), ENCCs display differential enteric progenitor gene expression patterns (*Taylor et al., 2016*) and neurochemical signatures representative of varying stages of neuronal differentiation and subtype diversification (*Poon et al., 2003*; *Holmqvist et al., 2004*; *Uyttebroek et al., 2010*). Zebrafish early differentiating enteric neurons have been characterized by the mRNA expression of *sox10*, *phox2bb*, *gfra1a*, *fgf13b*, and *ret*, as well as the immunoreactivity of Elavl3/4 (*Shepherd et al., 2004*; *Heanue and Pachnis, 2008*; *Uyttebroek et al., 2010*; *Taylor et al., 2016*). In addition, at this time, enteric neurons express multiple neurochemical markers, with Nos1 being most prominent (*Olden et al., 2008*; *Uyttebroek et al., 2010*), a finding consistent with studies performed within the amniote ENS (*Hao and Young, 2009*; *Matini et al., 1995*; *Qu et al., 2008*; *Heanue et al., 2016*). In light of these previous observations, our 68–70 hpf dataset was expected to contain the transcriptomes of ENCCs captured at various stages of their progressive differentiation into the diverse subtypes of the ENS.

We identified differentiating enteric neurons within the 68–70 hpf dataset based on the combinatorial expression of *elavl3*, *phox2bb*, *ret*, and *gfra1a* (*Figure 4G*), which mapped to the neural/neuronal major cell type regions of the dataset (*Figure 1E*), comprising Clusters 5 and 12 (*Figure 1E*). Transcripts that encode for the neurochemical marker *nos1*, and the neuropeptides *vip* and *vipb*, a paralogue to *vip* (*Gaudet et al., 2011*), were found in a subpopulation of enteric neurons localized to a distal group of the neuronal cluster, likely indicative of a differentiating enteric neuron subtype (*Figure 4G*; red arrows). We then queried for the presence of a combination of pan-neuronal and enteric neuron markers (*Figure 4H*). The pan-neuronal markers *tuba2*, *elavl3*, *stx1b*, and *gng2/3* (*Asakawa and Kawakami, 2010*; *Kelly et al., 2008*; *Park et al., 2000*; *Zheng et al., 2018*), as well as the autonomic neuron markers, *phox2a* and *phox2bb* (*Hans et al., 2013*), were present in both Clusters 5 and 12 (*Figure 4H*). However, the enteric neuron markers, *gfra1a*, *ret*, *hoxb5b*, *ngfrb*, *fgf13b*, *nos1*, *vipb*, and *vip* were mostly confined to Cluster 12, suggesting that this cluster contained differentiating enteric neurons (*Figure 4H*). Indeed, whole mount HCR analysis validated the spatiotemporal co-expression of *phox2bb*, *nos1*, *vipb*, and *elavl3* transcripts throughout the foregut of the zebrafish embryo by 69 hpf (*Figure 4I*; yellow arrowheads). These results suggest that *elavl3*[+]/*phox2bb*[+] early differentiating enteric neurons are first seen in the foregut and display an inhibitory neurochemical gene signature, consistent with prior observations in zebrafish and mammalian ENS (*Olden et al., 2008*; *Hao and Young, 2009*).

In an effort to examine the enteric neuron populations with finer resolution, Clusters 5 and 12 were subset from the main dataset in Seurat, re-clustered and visualized using a tSNE plot, producing 5 Sub-clusters (*Figure 5A*). The gene markers from each new Sub-cluster are provided in *Figure 5—source data 1*. The previously mentioned enteric neuron markers, with the addition of *etv1*, a recently identified marker of an enteric sensory neuron type, intrinsic primary afferent neurons (IPANs) in mouse (*Morarach et al., 2021*), were queried and visualized using dot and feature plot allowing the identification of Sub-cluster 3 as a differentiated enteric neuron cluster (*Figure 5A–C*). *nos1*, *vip*, and *vipb* were enriched in Sub-cluster 3 (*Figure 5B,C*; *Figure 5—figure supplement 1*). Interestingly, while expressed at lower average levels than in Sub-cluster 3, the enteric combination markers were also present in Sub-cluster 1 (*Figure 5B–C*). Sub-cluster 1 formed a central point from which Sub-cluster 3 could be seen emanating as a distal population (*Figure 5A*). Sub-clusters 1 and 3 likely depict enteric neurons captured at different stages along their progressive differentiation.

Given our hypothesis that the enteric neurons further along a differentiation program were localized to the distal tip of Sub-cluster 3, we asked whether this population of cells contained additional neurochemical or neuron subtype-specific differentiation genes. Within a small pocket of cells in Sub-cluster 3, we detected the expression of *calb2a* and *pbx3b* (*Figure 5D,E*; *Figure 5—figure supplement 1*), orthologous genes to *Calb2* and *Pbx3* that have previously been shown to denote adult myenteric IPANs in mammals (*Furness et al., 2004*; *Memic et al., 2018*), as well as the two acetylcholine associated genes, *acetylcholine esterase* (*ache, Bertrand et al., 2001*; *Huang et al., 2019*) and *vesicular acetylcholine transferase* (*slc18a3a, Hong et al., 2013*; *Zoli and Berlin, 2000*; *Figure 5E*; red arrowheads). Recently, a scRNA-seq study performed in E15.5 mouse demonstrated the co-expression of *Calb2*, *Pbx3*, and *Slc18a* during ENS development (*Morarach et al., 2021*). When examining IPAN gene markers in zebrafish larvae, HCR analysis revealed co-expression of *pbx3b*, *calb2a*, *vipb*, and *nos1* (*Figure 5G*; white arrowheads), or co-expression of *slc18a3a*, *calb2a*, *vipb*, and *nos1* (*Figure 5G*; yellow arrowheads), in discrete differentiating enteric neurons within the foregut region of the zebrafish gut at 68 hpf. These data indicate that zebrafish differentiating enteric neurons express IPAN gene signatures during their development. Collectively, our observations suggest that the *sox10*:GFP⁺68–70 hpf dataset captured an emerging IPAN population during its transition, where both excitatory and inhibitory neurochemical markers were co-expressed. Therefore, our single-cell analysis in zebrafish suggests that the transcriptional emergence of specific enteric neuron subtypes may be conserved between vertebrate species.

In order to identify novel signaling pathways within the developing enteric neuron population, the significantly enriched gene list from Sub-cluster 3 was processed using gene ontology (GO) pathway enrichment analysis. We found that three opioid signaling pathways were among the top 10 highest fold enriched pathways (*Figure 5—source data 2*; *Mi et al., 2019*). These pathways contained the G-protein-coupled receptors, *oprl1* and *oprd1b*, respectively representing nociception/orphan FQ (N/OFQ) peptide (NOP)-receptor and delta-opioid receptor (DOR) subtype members within the opioid receptor superfamily (*Figure 5H*; *Donica et al., 2013*; *Sobczak et al., 2014*). Specifically, feature plots showed that expression of the opioid receptors was tightly confined within the pocket of enteric neuron progenitors we identified as undergoing sensory lineage-specification, suggesting the expression of opioid receptor genes within both excitatory and inhibitory neurons at the early stages of enteric neuron differentiation within the zebrafish ENS (*Figure 5E and H*). Confirming this suggestion using HCR analysis, we observed the combinatorial expression of *oprl1*, *vipb*, and *phox2bb* within migrating enteric neuron progenitors at 68 hpf along the foregut (*Figure 5I–N*). The presence of opioid receptors within immature enteric neurons undergoing lineage-specification helps us to better understand the complexity of early ENS signaling and highlights an area that requires further investigation.

Based on our observation that the enteric neuron population that comprised Sub-cluster 3 only made up one of five *phox2bb+* Sub-clusters (*Figure 5A–C*), we suspected that the remaining Sub-clusters were made up of closely related autonomic neurons. In order to better visualize specific differences between the Sub-clusters, we viewed them using UMAP (*Becht et al., 2019*; *Mcinnes et al., 2018*; *Figure 5—figure supplement 2*). While the identity of the previous tSNE Sub-clusters were maintained, UMAP analysis allowed us to better visualize the separation between the Sub-clusters, which we were able to broadly classify as autonomic neurons based on their shared expression of *ascl1a*, *hand2*, *phox2a*, and *phox2bb* (*Figure 5—figure supplement 2A,B,D*). Within the population of autonomic neurons, we were able to distinguish a population of sympathetic neurons within Sub-cluster 2 based on their combinatorial expression of *th*, *dbh*, *lmo1*, and *insm1a* (*Figure 5—figure supplement 2B,E*), which could be clearly distinguished from the enteric neuron Sub-cluster 3 (*Figure 5—figure supplement 2B,F*). Using a cluster tree, we were able to confirm the distinction between enteric neuron Sub-Cluster three and sympathetic neuron Sub-cluster 2 (*Figure 5—figure supplement 2C*). Taking together the architecture of the UMAP clusters and their respective gene expression signatures, Sub-cluster 0 appears as a common sympatho-enteric neuron pool that lacks the expression of sympathetic and enteric specific neurochemical markers (*Figure 5—figure supplement 2A,B*). Sub-clusters 1 and 4 emanate as two distinct populations from Sub-cluster 0 and respectively exhibit lower expression levels of sympathetic and enteric markers comparative to the sympathetic and enteric neuron Sub-clusters 2 and 3 (*Figure 5—figure supplement 2A,B,E,F*). Overall, these results suggest that Sub-cluster 0 cells may represent a pool of immature sympatho-enteric neurons, and that Sub-clusters 1 and 4 both represent further differentiated, yet still

immature, pools of enteric and sympathetic neurons captured during the process of lineage specification into their respective terminal enteric and sympathetic neuron populations represented in Subclusters 3 and 2.

## Atlas of sox10:GFP⁺ cell types encompassing the embryonic to larval transition

To describe the dynamic transcriptional relationship between *sox10*:GFP⁺ cells across both time points, we merged the 48–50 hpf and 68–70 hpf datasets using Seurat's dataset and Integration and Label Transfer utility (*Stuart et al., 2019*). The merged datasets were visualized via UMAP, where we detected 27 clusters (*Figure 6—figure supplement 1A*). We observed that every cluster identified in the 48–50 hpf dataset mapped proximally to clusters at 68–70 hpf (*Figure 6—figure supplement 2A*). We labeled each cell in the UMAP using the previously described major cell type categories (*Figure 1*) forming a major cell type atlas (*Figure 6—figure supplement 1B*). Further refinement of the cell identities based on our previous annotations (*Figure 1—source data 2*) allowed us to form a higher resolution atlas for each cell type (*Figure 6A*). The top significantly enriched markers for each major cell type in the atlas are provided in a table in *Figure 6—source data 1*.

The atlas revealed transcriptionally similar populations among posterior *sox10*:GFP⁺ cells across the embryonic to larval transition. Illustrating this point, 48h-Cluster 2 and 68h-Cluster 8 both showed a high degree of similarity, as well as consistent *barx1, dlx2a,* and *twist1a* expression (*Figure 6B*), consistent with our prior analysis (*Figure 3C,D*). Furthermore, the central node of the pigment region within the atlas was marked by 48h-Cluster 8, which resolved into respective pigment chromatophore clusters at 68–70 hpf (*Figure 6C*). Specifically, we observed the early specified melanophore population at 48h-Cluster 8 branched into later stage melanophore populations (68h-Clusters 4 and 18). Further, we observed that the common bi-potent pigment progenitor population (68h-Cluster 13) bridged both melanophore clusters and the iridophore 68h-Cluster 16.

Cells within the neural/neuronal clusters assembled such that progenitor cells bridged into differentiating neurons spatially from the top to the bottom of the neural/neuronal region of the atlas (*Figure 6D*). Six clusters (Clusters 0, 5, 7, 13, 15, 17) were represented from 48 to 50 hpf, and five clusters (Clusters 3, 5, 10, 12, 14) from 68 to 70 hpf. Of interest, the 68–70 hpf neural progenitor populations (Clusters 3 and 10) shared common gene expression with the 48–50 hpf NCC population (48h-Cluster 5), reflected largely by their co-expression of *sox10, notch1a, dla,* and *foxd3* (*Figure 6D*; *Figure 6—figure supplement 1D*). We confirmed the spatiotemporal expression domains of *notch1a* and *dla* along the hindbrain, spinal cord, and in NCC populations along the post-otic vagal domain at 48 hpf (*Figure 3—figure supplement 1E,F*; arrowheads), in particular with *dla* in the ENCCs along the foregut (*Figure 3—figure supplement 1F*; arrow), a pattern similar to the ENCC makers *sox10* and *phox2bb* (*Figure 3—figure supplement 1A,B*). Delineated from the neural progenitor cells, we observed a bifurcation in cell states; with one moving toward a Schwann/glial cell fate, while the other branched toward neuronal. The glial arm followed a temporal progression of earlier cell fates at 48–50 hpf (48h-Cluster 15) toward the more mature fates at 68–70 hpf (68h-Cluster 14), both denoted by expression of *olig2* and *pou3f1*, respectively (*Figure 6D*). From 48h-Cluster 13, we observed the beginning of the neuronal populations, namely 48h-Clusters 0, 7, 13, and 17 and 68h-Clusters 5 and 12. The neuronal progenitor clusters (48h-Clusters 0, 13, and 17; 68h-Cluster 5) formed a spectrum of cell states leading toward the more mature neuronal populations (48h-Cluster 7; 68h-Cluster 12). For example, enteric progenitors culminated into a pool of enteric neurons, with the specific neural signature: *vipb, nos1, gfra1a, fgf13b,* and *etv1* (*Figure 6D*; *Figure 6—figure supplement 1D*).

Closer inspection of the pigment, mesenchymal, and neural/neuronal clusters separated by time highlighted both predicted and novel changes in gene expression patterns (*Figure 6—figure supplement 2*). For example, while the xanthophore differentiation marker *xdh* demonstrated expected restriction in expression to 68–70 hpf, we identified genes with no known roles in pigment cell development differentially expressed between the two stages, such as *rgs16* and *SMIM18* (*Figure 6—figure supplement 2B*). Moreover, within the mesenchyme lineages, both *barx1* and *snai1b* followed expected temporal expression trends, while *abracl* and *id1* both demonstrated novel differential gene expression profiles within the mesenchyme (*Figure 6—figure supplement 2C*). Lastly, the neural/neuronal lineage showed expected differential gene expression of genes such as *etv1* and *vipb*

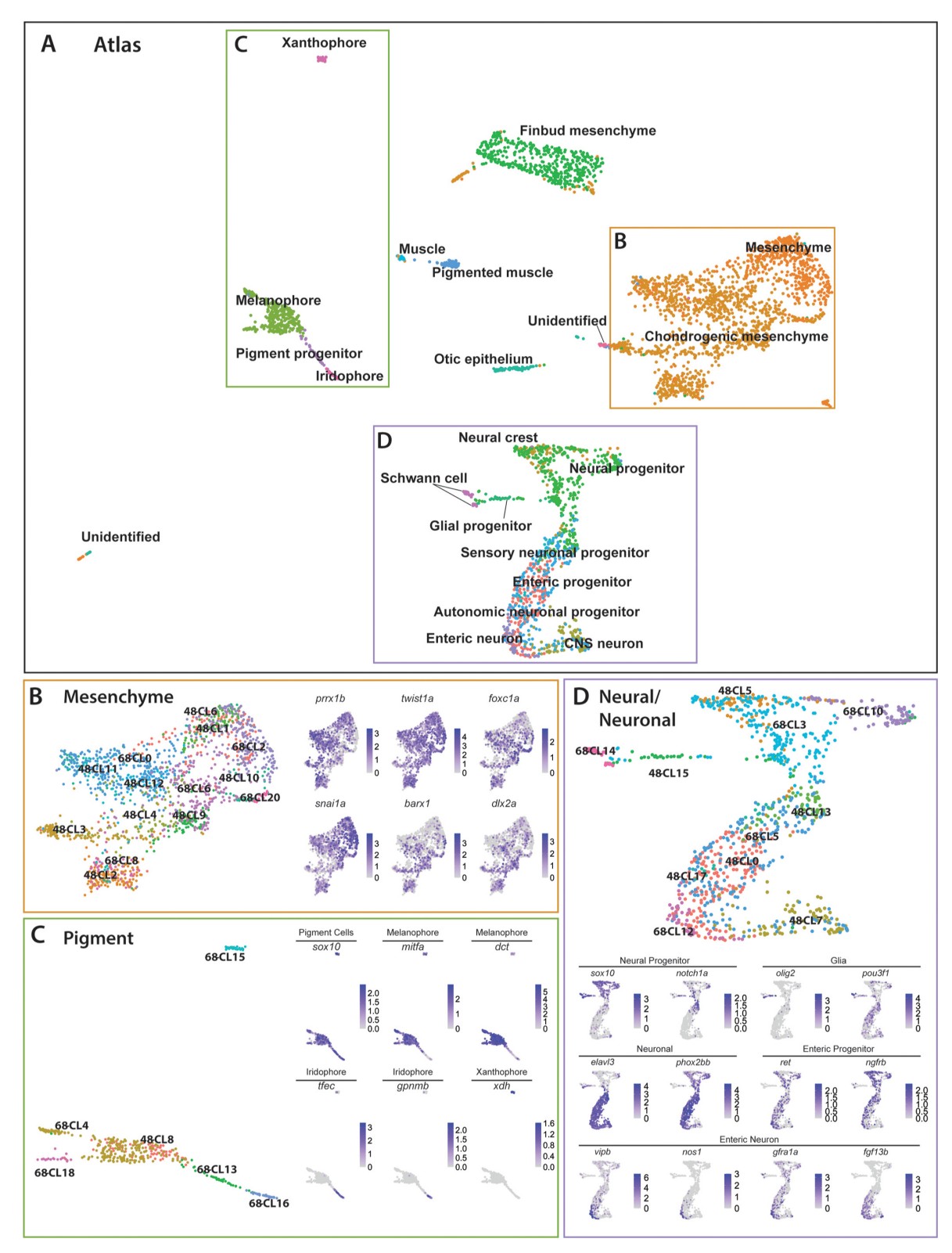

**Figure 6.** Integrated atlas of posterior *sox10*:GFP⁺ cell types spanning the embryonic to larval transition. (**A**) Global UMAP embedding demonstrating the clustering of cell types across 48–50 hpf and 68–70 hpf. Cell labels were transferred from the original curation (***Figure 1—source data 2***) to the new atlas after its creation, allowing for assessment of cell type organization. (**B**) Previously identified mesenchyme clusters form a large discernible cluster marked by *prrx1b*, *twist1a*, *foxc1a*, and *snai1a*, which was separated into both chondrogenic and general mesenchyme, as denoted by its differential
*Figure 6 continued on next page*

*Figure 6 continued*

expression of *barx1* and *dlx2a*. Importantly, nearly every 48–50 hpf cell type nests with a cluster at 68–70 hpf. (**C**) Pigment cells clusters reflect differentiation paths described in *Figure 4A*. Melanophores at 48–50 hpf group near to the 68–70 hpf melanophore cluster, bipotent pigment progenitors bridges both the iridophores and melanophores. Xanthophores cluster separately, reflecting their distinct lineage of origin at this developmental window. (**D**) Detailed analysis of the larger neural/neuronal cluster shows clear progression of cell fates from progenitor to differentiating glia or neuron. The expression of enteric neuronal markers is distinct from other subtypes at this dataset.

The online version of this article includes the following source data and figure supplement(s) for figure 6:

**Source data 1.** List of marker genes per major cell type identity in the *sox10*:GFP⁺ merged atlas.
**Figure supplement 1.** Annotated *sox10*:GFP⁺ atlas labeled by cell types.
**Figure supplement 2.** Differential expression among pigment, mesenchyme and neural/neuronal subsets of the *sox10*:GFP⁺ atlas (**A**) UMAP visualization of cells labeled by source identity (either 48–50 hpf or 68–70 hpf) following integration.

at 68–70 hpf, while revealing novel expression of *nova2* and *zgc:162730,* which have previously uncharacterized roles in *sox10*-derived cells (*Figure 6—figure supplement 2D*). Together, these findings demonstrate dynamic gene expression changes across developmental stages occurring among s*ox10*:GFP⁺ cells and highlights novel genes for further study.

## A hox gene signature within sox10-derived cells in the posterior zebrafish

A common theme examined by many recent and insightful single-cell profile studies of the NCC (*Dash and Trainor, 2020*; *Soldatov et al., 2019*) is that the expression of *hox* genes, which encode for Homeobox transcription factors, display discrete expression patterns between various cell lineages, such as in the cranial NCC. We wondered whether specific *hox* signatures were expressed within posterior NCC and their recent derivatives. To analyze if we could detect *hox* gene patterns within the atlas, we queried all the known canonical *hox* genes within zebrafish as listed on zfin.org (*Ruzicka et al., 2019*). We detected broad expression of 45 of the 49 zebrafish *hox* genes across the atlas, with 85% of the cells in the atlas expressing at least one *hox* gene (*Figure 7—figure supplement 1A,J*). The four undetected hox genes (*hoxc1a*, *hoxc12b*, *hoxa11a*, and *hoxa3a*) were not examined further.

A dot plot revealed that specific *hox* gene expression patterns demarcated distinct tissues, with specific robustness in the neural fated cells (*Figure 7A*). Common to the neural lineages, we observed a core *hox* profile which included *hoxb1b*, *hoxc1a*, *hoxb2a*, *hoxb3a*, *hoxc3a*, *hoxd3a*, *hoxa4a*, *hoxd4a*, *hoxb5a*, *hoxb5b*, *hoxc5a*, *hoxb6a*, *hoxb6b*, and *hoxb8a* (*Figure 7A*). One of the top expressed constituents of the core signature, *hoxa4a*, was also widely expressed in several other lineages (*Figure 7A*; *Figure 7—figure supplement 1*). The *hox* signature applied to the NCC, neural progenitor, enteric progenitor, enteric neuron, glial progenitor, autonomic neuronal progenitor, and CNS lineages described in the atlas. Clustering of all the atlas lineages relying only on *hox* gene expression highlighted the robustness of the core *hox* signature to distinguish the neural lineage fates, grouping the differentiating (autonomic neuronal progenitors, enteric neurons, enteric progenitors and CNS neurons) and progenitor lineages (neural progenitors and glial progenitors) into neighboring clades (*Figure 7B*). Building on the core neural signature unifying the neural fates, slight variations in *hox* expression between autonomic and enteric lineages distinguished them from one another, which are summarized in *Figure 7E*. Most notably, considering the lineages in increasing specificity of cell fate, there was a detectable shift in *hox* expression among the autonomic neural progenitors to the enteric neuronal lineage, demarcated by the increase in *hoxb2a*, *hoxd4a*, *hoxa5a*, *hoxb5a*, and *hoxb5b,* accompanied by diminished expression of *hoxc3a*, *hoxc5a*, *hoxb6a* and *hoxb8a*, which formed a distinctive enteric hox signature (*Figure 7—figure supplement 1A,K–N*).

In order to better understand the complexities of *hox* codes within specific lineages, we performed a pairwise comparison of each *hox* gene for autonomic and enteric lineages, counting the number of cells which co-expressed each *hox* pair. Examining the autonomic neuronal progenitors (*Figure 7C*) and the enteric neuron populations (*Figure 7D*), both lineages demonstrated pervasive fractions of co-positive cells for combinations of the core *hox* signature. For example, autonomic neuronal cells were enriched with a high fraction of pairwise combinations for *hoxc1a*, *hoxa4a*, *hoxb3a*, and/or *hoxb5b*(*Figure 7C*). The enteric signature was highly enriched in the unique

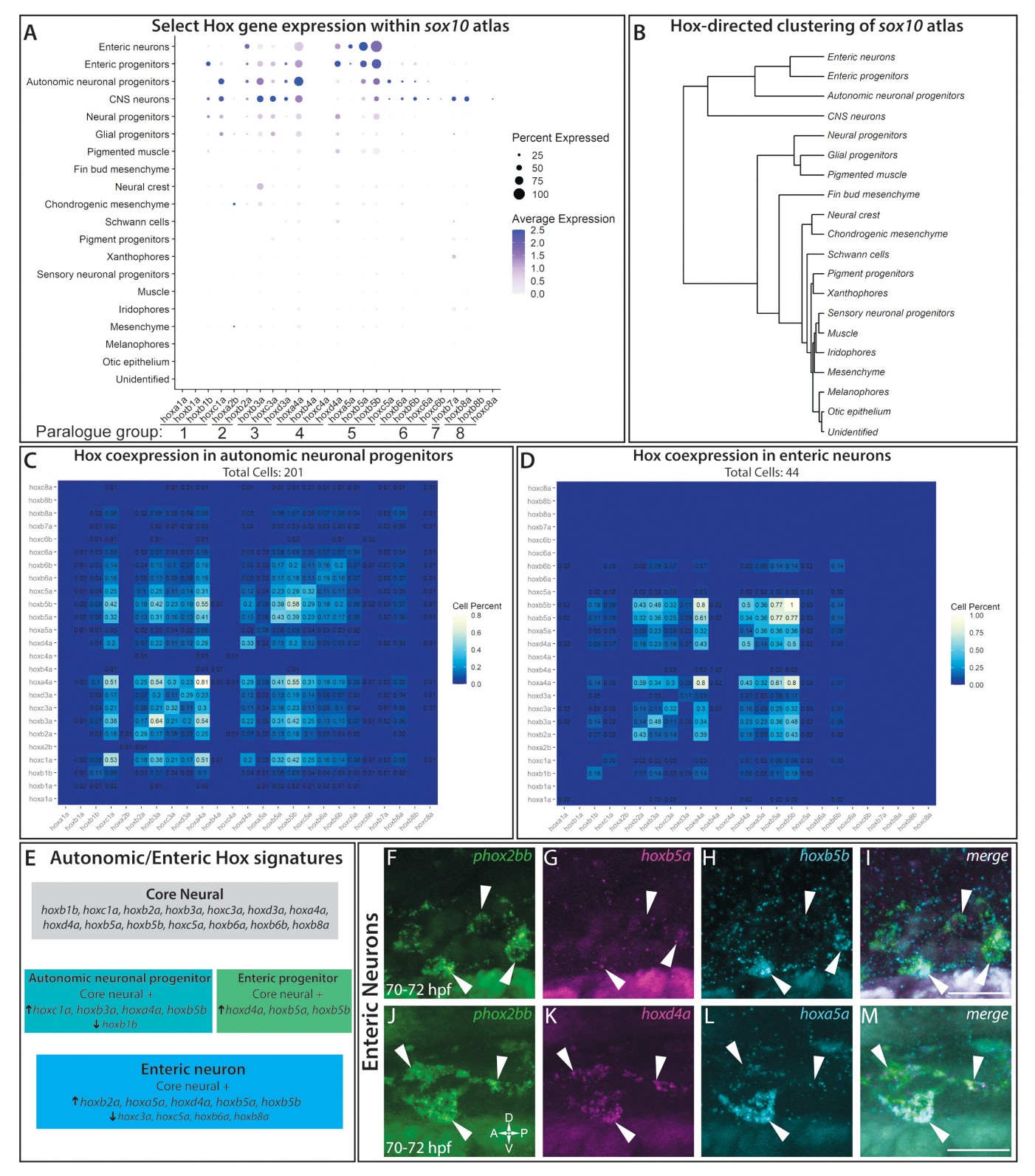

**Figure 7.** *hox* genes expressed across cell lineages within the *sox10*:GFP[+] atlas. (**A**) Dot plot shows both the mean expression (color) as well as percent of cells (size) per lineage for zebrafish *hox* genes in the first eight paralogy groups (PG). The full list of *hox* gene expression profiles per lineage can be found in **Figure 7—figure supplement 1**. Discrete *hox* profiles discern specific cell types, which is particularly evident in the enteric neuronal cluster. (**B**) Clustering of atlas lineages based *hox* expression profiles groups highlights robust core neural signature, which distinguishes the neural lineages

*Figure 7 continued on next page*

*Figure 7 continued*

from the remainder of the clades. Neural and glial progenitors formed an intermediate clade between the low-*hox* expressing lineages and the main neural branch. Additionally, the fin bud mesenchyme, which also has a highly distinctive *hox* profile, also forms a distinct clade. Subtle variations in *hox* expression by remaining lineages are further reflected in the remaining portion of the dendrogram; however, these are far less distinct. (C–D) Pairwise comparison of the fraction of cells in either the autonomic neural progenitor lineage (C) or the enteric neurons (D) for the first eight parology groups. Intersection of the gene pairs reflect the fraction of cells with expression for both genes with a log2 Fold change values > 0, with the identical gene intersections along the primary diagonal representing the total number of cells which express that gene in the lineage. Enteric neural *hox* signature was not only specific to this cell population, but also was abundantly co-expressed. (E) Summary panel describing the specific autonomic and enteric *hox* signatures detected. A common *hox* expression profile, referred to as the core signature, was found that is then modified across the specific lineages. (F–M) In situ validation of the chief enteric neural *hox* signature via HCR. *phox2bb* (F–J) labels enterically fated neurons at the level of the midgut in larval stage embryos fixed at 70–72hpf. White arrows highlight specific cells of interest. Key hox signature constituents *hoxb5a* (G) and *hoxb5b* (H) or *hoxd4a* (K) and *hoxa5a* (L) were found to be co-expressed within *phox2bb* expressing cells (White Arrows). Scale bars in (I,M): 50 μm.

The online version of this article includes the following figure supplement(s) for figure 7:

**Figure supplement 1.** Comprehensive overview of *hox* expression profiles within the atlas.

expression of *hoxa5a*, with co-expression for *hoxb5a* (36%) and *hoxb5b* (36%), as well as strong co-expression of *hoxd4a* or *hoxb5a* with *hoxb5b* (*Figure 7D*).

To confirm that *hox* core genes were co-expressed within enteric neurons, we sought to validate their expression patterns using HCR probes. As previously described, *hoxd4a*, *hoxa5a*, *hoxb5a*, and *hoxb5b* all exhibited strong hindbrain expression (*Figure 7—figure supplement 1B–I*; *Barsh et al., 2017*), confirming the specificity of our probes. At 70–72 hpf, as predicted by our analysis, enteric neurons along the level of the midgut, marked by *phox2bb* expression (*Figure 7F,J*), were *hoxb5a$^+$/hoxb5b$^+$* (*Figure 7G–I*) and *hoxd4a$^+$/hoxa5a$^+$* (*Figure 7K–M*). These data confirm that enteric neurons co-express enteric *hox* code genes during their early development.

With respect to the remaining cluster identities (*Figure 7A*), many of the populations showed varied *hox* expression profiles. Both the chondrogenic and general mesenchyme clusters demonstrated *hoxa2b* expression, as well as weak expression for *hoxb2a*, *hoxb3a*, and *hoxd4a*. Our detection of these *hox* expression profiles was consistent with prior reports that they are expressed within NCC targets toward the posterior pharyngeal arches, as well as migrating NCC (*Minoux and Rijli, 2010*; *Parker et al., 2018*; *Parker et al., 2019*). We detected the distinct identity of the fin bud mesenchyme (*Ahn and Ho, 2008*; *Nakamura et al., 2016*) through the expression of *hoxa9b*, *hoxa10b*, *hoxa11b*, *hoxa13b*, *hoxd9a*, and *hoxd12a* (*Figure 7A*; *Figure 7—figure supplement 1P–Q*). The pigment populations, including the pigment progenitors, melanophores, iridophores, and xanthophores, contained generally low levels of *hox* gene expression. Despite this, we still observed a slight variation of *hox* expression among the pigment populations. For example, low levels of *hoxa4a*, *hoxb7a*, *hoxb8a*, *hoxc3a*, and *hoxd4a* were detected among the iridophore population, while only *hoxb7a* was detected within a high fraction of xanthophores (*Figure 7—figure supplement 1A*). Interestingly, these expression profiles are not shared by the melanophore population, which displayed uniformly very low levels of detectable *hox* expression. Lastly, the muscle, otic, and unidentified cells showed almost no *hox* expression profile, which serves a foil for the specificity of the signatures outlined. We noted that the 'pigmented muscle' cluster weakly mirrored the general neural *hox* signature, likely a shared signature more reflective of the axial position of the muscle cells rather than a shared genetic profile, as corroborated by their distinct separation of the clusters on the atlas UMAP (*Figure 6A*).

Overall, these above described *hox* signatures detected within our scRNA-seq atlas indicates that distinct cell types express unique *hox* combinations during their delineation. Description of the *hox* signatures within the *sox10* atlas provides further tools to identify these discrete cell populations, as well as exciting new avenues for further mechanistic investigation.

## Discussion

We present a single-cell transcriptomic atlas resource capturing diversity of posterior-residing *sox10*-derived cells during the embryonic (48–50 hpf) to early larval transition (68–70 hpf) in zebrafish. We identified a large number of cell types; including pigment progenitor cells delineating into distinct chromatophores, NCC, glial, neural, neuronal, otic vesicle cells, muscle, as well as transcriptionally

distinct mesenchymal cell populations, extending prior whole embryo-based zebrafish single cell studies (*Farnsworth et al., 2020*) and expanding the resolution at which these cells have been described to date. Our study is the first single-cell transcriptomic analysis covering early ENS development in zebrafish, in addition to analysis of posterior *sox10⁺* mesenchyme and pigment cells present during the late embryonic to larval phase. The developmental window we examined, the embryonic to larval transition, is regarded as an ephemeral phase (*Singleman and Holtzman, 2014*) and as such is expected to contain the dynamic cell differentiation states that we observed within our atlas. We discovered that distinct *hox* transcriptional codes demarcate differentiating neural and neuronal populations, highlighting their potential roles during cell subtype specification. We uncovered evolutionarily conserved and novel transcriptional signatures of differentiating enteric neuron cell types, thereby expanding our knowledge of ENS development. Corroborating our transcriptomic characterizations, we validated the spatiotemporal expression of several key cell type markers using HCR. Collectively, this comprehensive cell type atlas can be used by the wider scientific community as a valuable resource for further mechanistic and evolutionary investigation of posterior *sox10*-expressing cells during development and the ontogenesis of neurocristopathies. The atlas is available via an interactive cell browser (https://zebrafish-neural-crest-atlas.cells.ucsc.edu/).

Collectively, our single-cell datasets captured the transition from enteric neural progenitor to differentiating enteric neuron subtype (*Figures 4*, *5* and *6*). A similar enteric population consisting of *Sox10*, *Ret*, *Phox2b*, and *Elavl4* was identified by scRNA-seq in the mouse (*Lasrado et al., 2017*), indicating zebrafish express conserved enteric programs. As well, a recent scRNA-seq study performed using E15.5 mice, a time point further along in ENS development when compared to our zebrafish study described here, suggests that *Nos1⁺/Vip⁺* cells represent a post-mitotic immature neuron population capable of branching into excitatory and inhibitory neurons via subsequent differentiation mediated by lineage-restricted gene expression (*Morarach et al., 2021*). Their model posits that *Nos1⁺/Vip⁺/Gal⁺* enteric neurons are capable of assuming an IPAN signature, characterized by the loss of *Vip* and *Nos1*, and the gain of *Calb*, *Slc18a2/3*, and *Ntng1*; a process regulated by transcription factors, *Pbx3* and *Etv1*. This model of IPAN formation appears congruent with a previous birth dating study performed in mice, where researchers demonstrated the transient expression of *Nos1* in enteric neurons (*Bergner et al., 2014*). Our observations in zebrafish (*Figure 5*; *Figure 5—figure supplement 1*; *Figure 6*) corroborates the proposed mammalian gene expression model and suggests that we captured a transitional time point where subsequent differentiation is just being initiated (*Figure 5F*) and suggests an evolutionarily conserved mechanism of ENS formation across vertebrate species.

Within the enteric neuron subpopulation in our dataset, we discovered enrichment of the opioid pathway genes *oprl1* and *oprd1b*, respectively encoding for NOP and DOR class opioid receptors (*Figure 5*; *Donica et al., 2013*; *Holzer, 2004*). The expression of opioid receptors has previously been shown in inhibitory interneurons found within the adult ENS where they are known to play functional roles during gut homeostasis (*DiCello et al., 2020*; *Lay et al., 2016*; *Wood and Galligan, 2004*). While the presence and inhibitory effect of opioid receptors is well characterized within the adult ENS, the role of opioid signaling during the earliest stages of enteric neuron maturation and ENS formation has yet to be investigated. Indeed, a recent study performed in zebrafish found that *oprl1* was expressed within 7 dpf enteric neurons following bulk RNA sequencing (*Roy-Carson et al., 2017*). However, these data represent a developmental stage where zebrafish are characterized as free-swimming larvae that display feeding behavior and digestive capability representative of a functioning and more mature ENS (*Cassar et al., 2018*). As such, our 68–70 hpf dataset, in which we detected the expression of *oprl1* and *oprd1b*, represents the earliest stage in which they have been shown to be expressed within the early developing ENS. The presence of opioid receptor transcripts within the ENS at 68–70 hpf, a time when immature enteric neurons are continuing to migrate and pattern within the developing embryo, highlights an important area of research focusing on the interplay between opioid use and fetal ENS development, a reality that has been on the rise in recent decades as opioid abuse continues to increase.

Finally, we have elucidated a comprehensive, combinatorial code of *hox* expression which defines specific cell lineages within the context of the *sox10* atlas. The fin bud mesenchyme, pigment populations, and neural fates presented the most specific *hox* codes (*Figure 7*). Among the neural lineages, we identified a previously undescribed *hox* signature demarcating the developing enteric neuron population in zebrafish: high relative expression of *hoxb2a*, *hoxa5a*, *hoxd4a*, *hoxb5a*, and

*hoxb5b,* while also exhibiting low expression of *hoxc3a, hoxc5a, hoxb6a,* and *hoxb8a* (*Figure 7*). While previous bulk microarray studies of differentiating enteric neurons from humans and mice have been shown to express orthologs to the former signature (*Heanue and Pachnis, 2008*; *Memic et al., 2018*), our analysis extends knowledge to comprehensively show for the first time specific combinatorial *hox* expression at a single-cell resolution, thereby providing a heretofore unknown readout of *hox* heterogeneity among nascent enteric cells. The conservation of an enteric *hox* signature between zebrafish and other systems points toward a larger conservation of function, which may facilitate translation of future findings between models. These findings imply interesting potential hypotheses wherein combinations of *hox* codes may represent a molecular address designating the axial site of origination for migrating NCC or indicate dynamic expression profiles which are modified during the NCC to enteric neuron developmental course. Further work is required to test these possible models as well as the functional requirement of the constituent members of the *hox* code in the developing zebrafish ENS. Considered collectively, these data lend support to a model in which overlapping expression domains of *hox* genes may facilitate enteric neural subtype differentiation, similar to their function in hindbrain and spinal neurons (*Philippidou and Dasen, 2013*).

In summary, our study greatly increases foundational understanding of NCC-derived cell fates, as well as other *sox10*$^+$posterior cell types in zebrafish, thereby complementing ongoing studies in mammalian models and expanding fundamental knowledge of how cells diversify in developing organisms. The spatiotemporal information contained within our zebrafish atlas will serve as a resource for the developmental biology, stem cell, evolutionary biology and organogenesis communities.

# Materials and methods

## Key resources table

| Reagent type (species) or resource | Designation | Source or reference | Identifiers | Additional information |
|---|---|---|---|---|
| Recombinant DNA reagent | *phox2bb* | *Uribe and Bronner, 2015* | | |
| Recombinant DNA reagent | *sox10* | *Dutton et al., 2001* | | |
| Recombinant DNA reagent | *mmp2* | | *Mammalian Gene Collection Program Team et al., 2002* | |
| Sequence-based reagent | *notch1a* | Uribe Lab | | Forward 5′-CAG TGGAC TCAGCAGCA TC-3′ Reverse 5′-CCTTCCGAC-CAATCAGA-CAAG-3′ |
| Sequence-based reagent | *dla* | Uribe Lab | | Forward 5′-CAGCCAAG TTGCTCAGAG-3′ Reverse 5′-G TACAGAGAAC-CAGCTCATC-3′ |
| Sequence-based reagent | *foxc1a* | Uribe Lab | | Forward 5′-A TACGGTGGAC TCTGTGG-3′ Reverse 5′-CAGCGTCTG TCAGTATCG-3′ |
| Genetic reagent (*Danio rerio*) | AB | ZIRC | | Wild-type zebrafish |

*Continued on next page*

*Continued*

| Reagent type (species) or resource | Designation | Source or reference | Identifiers | Additional information |
|---|---|---|---|---|
| Genetic reagent (*Danio rerio*) | Tg(−4.9sox10: egfp)ba2Tg | *Carney et al., 2006* | | GFP Labeled Neural Crest Cells |
| Commercial assay, kit | *mitfa* | Molecular Instruments | NM_130923.2 | |
| Commercial assay, kit | *tfec* | Molecular Instruments | NM_001030105.2 | |
| Commercial assay, kit | *xdh* | Molecular Instruments | XM_683891.7 | |
| Commercial assay, kit | *phox2bb* | Molecular Instruments | NM_001014818.1 | |
| Commercial assay, kit | *ngfrb* | Molecular Instruments | NM_001198660.1 | |
| Commercial assay, kit | *gfra1a* | Molecular Instruments | NM_131730.1 | |
| Commercial assay, kit | *crestin* | Molecular Instruments | AF195881.1 | |
| Commercial assay, kit | *foxd3* | Molecular Instruments | NM_131290.2 | |
| Commercial assay, kit | *vipb* | Molecular Instruments | NM_001114555.1 | |
| Commercial assay, kit | *elavl3* | Molecular Instruments | NM_131449 | |
| Commercial assay, kit | *oprl1* | Molecular Instruments | NM_205589.2 | |
| Commercial assay, kit | *barx1* | Molecular Instruments | NM_001024949.1 | |
| Commercial assay, kit | *pbx3b* | Molecular Instruments | BC131865.1 | |
| Commercial assay, kit | *prrx1b* | Molecular Instruments | NM_200050.1 | |
| Commercial assay, kit | *slc18a3a* | Molecular Instruments | NM_0010775550.1 | |
| Commercial assay, kit | *hoxb5a* | Molecular Instruments | NM_131101.2 | |
| Commercial assay, kit | *hoxb5b* | Molecular Instruments | bc078285.1 | |
| Commercial assay, kit | *hoxd4a* | Molecular Instruments | NM_001126445 | |
| Commercial assay, kit | *hoxa5a* | Molecular Instruments | NM_131540.1 | |
| Commercial assay, kit | *twist1a* | Molecular Instruments | NM_130984.2 | |
| Commercial assay, kit | Single Cell 3' v2 Chemistry Kit for 10,000 cells | 10x Genomics | CG00052 | https://support. 10xgenomics. com/single-cell-gene-expression/ library-prep/ doc/user-guide-chromium-single-cell-3-reagent-kits-user-guide-v2-chemistry |

*Continued on next page*

*Continued*

| Reagent type (species) or resource | Designation | Source or reference | Identifiers | Additional information |
|---|---|---|---|---|
| Commercial assay, kit | NextSeq 500/550 Mid Output Kit v2.5 (150 Cycles) | illumina | 20024904 | https://www.illumina.com/products/by-type/sequencing-kits/cluster-gen-sequencing-reagents/nextseq-series-kits-v2-5.html |
| Chemical compound, drug | 1-phenyl 2-thiourea (PTU)/ E3 solution | Karlsson 741 et al., 2001 | Sigma-Aldrich | P7629 |
| Chemical compound, drug | Tricaine | A5040 | Sigma | |
| Chemical compound, drug | Accumax buffer | A7089 | Sigma-Aldrich | |
| Chemical compound, drug | Hank's Buffer | 55021C | Sigma-Aldrich | |
| Chemical compound, drug | Phusion-HF | M0530S | New England Biolabs | |
| Chemical compound, drug | Zero Blunt TOPO PCR Cloning Kit | 451245 | Invitrogen | |
| Software, algorithm | R v3.6.3 | R-project | RRID:SCR_001905 | https://www.r-project.org/ |
| Software, algorithm | Seurat v3.1.1 | Satija Lab | RRID:SCR_007322 | https://github.com/satijalab/seurat |
| Software, algorithm | Fiji | PMID:22743772 | RRID:SCR_002285 | https://imagej.net/Fiji |
| Software, algorithm | IMARIS v9.2 | Bitplane | RRID:SCR_007370 | Bitplane.com |
| Software, algorithm | PANTHER | | GENEONTOLOGY Unifying Biology | RRID:SCR_004869 |
| http://pantherdb.org | | | | |
| Software, algorithm | Cell Ranger v2.1.0 | 10x Genomics | RRID:SCR_017344 | *Zheng et al., 2017*, https://support.10xgenomics.com/single-cell-gene-expression/software/pipelines/latest/what-is-cell-ranger |

## Animal husbandry, care, and synchronous embryo collection

Groups of at least 15 adult Tg(−*4.9sox10:*GFP)[ba2Tg] (*Carney et al., 2006*) zebrafish (*Danio rerio*) males and 15 females from different tank stocks were bred to generate synchronously staged embryos across several clutches. All embryos were cultured in standard E3 media until 24 hr post fertilization (hpf), then transferred to 0.003% 1-phenyl 2-thiourea (PTU)/E3 solution (*Karlsson et al., 2001*), to arrest melanin formation and enable ease of GFP sorting. While it has been suggested that high concentration (.03%) PTU incubation prior to 22 hpf may cause organism-wide effects,.003% PTU application after 22 hpf has been shown to have no major effects on NCC survival or pigment cell formation, as described (*Bohnsack et al., 2011*). Embryos were manually sorted for GFP expression and synchronously staged at 24 hpf. Care was taken such that embryos which exhibited developmental delay or other defects were removed prior to collection. All work was performed under protocols approved by, and in accordance with, the Rice University Institutional Animal Care and Use Committee (IACUC).

## Isolation of tissue and preparation of single-cell suspension

100 embryos between 48- 50 hpf and 100 larvae between 68–70 hpf were dechorionated manually and then transferred to 1X sterile filtered PBS, supplemented with 0.4% Tricaine (Sigma, A5040) to anesthetize. Tissue anterior to the otic vesicle and tissue immediately posterior to the anal vent was manually removed using fine forceps in 48–50 hpf embryos, while tissue anterior to the otic vesicle was removed from 68 to 70 hpf larvae, as schematized in *Figure 1*. This was to capture as many posterior *sox10*:GFP$^+$ cells in the later time point as possible. Remaining tissue segments were separated into nuclease-free tubes and kept on ice immediately following dissection. Dissections proceeded over the course of 1 hr. To serve as control for subsequent steps, similarly staged AB WT embryos were euthanized in tricaine and then transferred to sterile 1X PBS. All following steps were conducted rapidly in parallel to minimize damage to cells: Excess PBS was removed and tissue was digested in 30°C 1X Accumax buffer (Sigma-Aldrich, A7089) for 30–45 min to generate a single cell suspension for each sample. At 10 min intervals, tissue was gently manually disrupted with a sterile pipette tip. As soon as the tissue was fully suspended, the cell solutions were then transferred to a fresh chilled sterile conical tube and diluted 1:5 in ice cold Hank's Buffer (1x HBSS; 2.5 mg/mL BSA; 10 µM pH8 HEPES) to arrest the digestion. Cells were concentrated by centrifugation at 200 rcf for 10 min at 4°C. Supernatant was discarded carefully and cell pellets were resuspended in Hank's Buffer. Cell solution was passed through a 40 µm sterile cell strainer to remove any remaining undigested tissue and then centrifuged as above. Concentrated cells were resuspended in ice cold sterile 1X PBS and transferred to a tube suitable for FACS kept on ice. The 48–50 hpf and 68–70 hpf experiments were performed on completely separate dates and times using the above described procedures.

## Fluorescent cell sorting and single-cell sequencing

Fluorescent Assisted Cell Sorting (FACS) was performed under the guidance of the Cytometry and Cell Sorting Core at Baylor College of Medicine (Houston, TX) using a BD FACSAria II (BD Biosciences). Zebrafish cells sorted via GFP fluorescence excited by a 488 nm laser, relying on an 85 µm nozzle for cell selection. Detection of GFP$^+$ cells was calibrated against GFP$^-$ cells collected from AB wildtype embryos, as well as GFP$^+$ cells collected from the anterior portions of the *sox10*:GFP embryos. Optimal conditions for dissociated tissue inputs (number of embryos needed, etc.) and FACS gating was determined via pilot experiments prior to collection for subsequent scRNA-seq experiments. Sample preparation for scRNA-seq was performed by Advanced Technology Genomics Core (ATGC) at MD Anderson (Houston, TX). 4905 and 4669 FACS-isolated cells for the 48–50 and 68–70 hpf datasets were prepared on a 10X Genomics Chromium platform using 10X Single Cell 3' V2 chemistry kit for 10,000 cells. cDNA libraries were amplified and prepared according to the 10X Genomics recommended protocol, with details provided in *Figure 1—figure supplement 1C*. A 150 cycle Mid-Output flow cell was used for sequencing on an Illumina NextSeq500. Sequencing was aligned at MD Anderson ATGC to the DanioGRCz10 version of the zebrafish genome using the 10X Genomics Cell Ranger software (v2.1.0, *Zheng et al., 2017*). Gene reads per cell were stored in a matrix format for further analysis.

## Data processing and analysis

The 10x genomics sequencing data was then analyzed using Seurat (*Satija et al., 2015*; *Stuart et al., 2019*; *Butler et al., 2018*) v3.1.1 software package for R, v3.6.3 (*R Development Core Team, 2020*). The standard recommended workflow was followed for data processing. Briefly, for both the 48–50 hpf and 68–70 hpf datasets, cells which contained low (<200) or high (>2500) genes were removed from analysis. Gene expression was normalized using the NormilizeData command, opting for the LogNormalize method (Scale factor set at 10,000) and further centered using the ScaleData command. Variable features of the dataset were calculated with respect to groups of 2000 genes at a time. Both datasets were evaluated considering the first 20 principal components (PC) as determined by the RunPCA command with a resolution of 1.2 for PCA, tSNE, and UMAP analyses. The resolution value was empirically determined for the FindClusters by iterative assessment of cluster patterns generated values ranging from 0.4 to 2.0. The resolution was set to 1.2 which balanced between over segmentation of cell identities and effective separation of tSNE-mapped clusters. The

appropriate PCs were selected based on a Jack Straw analysis with a significance of p<0.01, as generated by the JackStraw command.

Clustering was performed using FindNeighbors and FindClusters in series. We identified 19 clusters in the 48–50 hpf dataset and 23 clusters in the 68–70 hpf dataset. Significantly enriched genes for each cluster were determined via a Wilcoxon Rank Sum test implemented by the FindAllMarkers command. From these expressed gene lists within each cluster, all cluster identities were manually curated via combinatorial expression analysis of published marker genes in the literature, zfin.org and/or bioinformatics GO term analysis via the Panther Database.

Generation of the merged atlas was performed via the FindIntegrationAnchors workflow provided in the Standard Workflow found on the Seurat Integration and Label Transfer vignette. Clustering was performed for the atlas based on the first 20 PCs, consistent with the original datasets. Subsets of the atlas discounted any spuriously sorted cells for clarity. All features plots represent expression values derived from the RNA assay. Sub-clustering of the enteric clusters was performed by sub-setting Clusters 5 and 12 from the 68 to 70 hpf dataset and reinitializing the Seurat workflow, as described above. Clusters were identified based on the first 6 PCs. Detection of cell cycle phase was conducted following the Cell cycle and scoring vignette. Genes used for identification of cell cycle phases can be found in *Figure 1—figure supplement 3*. Pairwise *hox* analysis was conducted using tools from the Seurat package in R by assessing the number of cells which had expression for *hox* gene pairs queried with a log2 fold change greater than 0. Dendrogram Cluster trees were generated using Seurat's BuildClusterTree function.

## Whole mount in situ hybridization

cDNAs for *foxc1a*, *notch1a*, and *dla* were amplified via high fidelity Phusion-HF PCR (NEB) from 48 hpf AB WT cDNA libraries using primers in the Key resources table. PCR products were cloned using the Zero Blunt TOPO PCR Cloning Kit (Invitrogen), as per manufacturer protocols, and sequenced validated. Plasmids encoding *phox2bb*, *sox10*, *mmp2* were generously sourced as listed in the key resources table. Antisense digoxigenin (DIG)-labeled riboprobes were produced from cDNA templates of each gene. AB wild type embryos were treated and stained to visualize expression as previously described in *Jowett and Lettice, 1994*. Following in situ reactions, embryos were post-fixed in 4% Paraformaldehyde (PFA) and mounted in 75% Glycerol for imaging. A Nikon Ni-Eclipse Motorized Fluorescent upright compound microscope with a 4X objective was used in combination with a DS-Fi3 color camera. Images were exported via Nikon Elements Image Analysis software.

## Whole mount hybridization chain reaction

HCR probes were purchased commercially (Molecular Instruments Inc, CA) and were targeted to specific genes based on their RefSeq ID (Key resources table). Whole mount HCR was performed according to the manufacturer's instructions (v3.0, *Choi et al., 2016*; *Choi et al., 2018*) on *sox10: GFP$^+$* or AB embryos previously fixed at the appropriate stage in 4% PFA.

## Confocal imaging and image processing

Prior to imaging, embryos were embedded in 1% low melt agarose (Sigma) and were then imaged using an Olympus FV3000 Laser Scanning Confocal, with a UCPlanFLN 20×/0.70NA objective. Confocal images were acquired using lambda scanning to separate the Alexafluor 488/Alexafluor 514 or the Alexafluor 546/Alexafluor 594 channels. Final images were combined in the FlowView software and exported for analysis in either Fiji (*Rueden et al., 2017*; *Schneider et al., 2012*; *Schindelin et al., 2012*) or IMARIS image analysis software (Bitplane). Figures were prepared in Adobe Photoshop and Illustrator software programs, with some cartoons created via https://biorender.com/.

## Data availability

The raw sequence read files and processed cellular barcode, gene, and matrix files produced by CellRanger are available in the National Center for Biotechnology Information's (NCBI) Gene Expression Omnibus (GEO) database (https://www.ncbi.nlm.nih.gov/geo/), accession number: GSE152906. The atlas/associated processed Seurat objects are available on the University of California, Santa Cruz (UCSC) Cell Browser (https://zebrafish-neural-crest-atlas.cells.ucsc.edu/). Code written for

Seurat data analysis is available on GitHub ([https://github.com/UribeLabRice/NeuralCrest_Atlas_2020](https://github.com/UribeLabRice/NeuralCrest_Atlas_2020); *UribeLabRice, 2021*; copy archived at swh:1:rev:195fe7dd0bf388ad4cd6f01db052f45d8b8fa67e).

## Acknowledgements

Funding for this project was provided by Rice University, Cancer Prevention and Research Institute of Texas (CPRIT) Recruitment of First-Time Tenure Track Faculty Members (CPRIT-RR170062) and the NSF CAREER Award (1942019) awarded to RAU, a Houston Livestock Show and Rodeo Research Award to JAM and PAB, and a SDB Choose Development! Fellowship award to JLW. We acknowledge the Cytometry and Cell Sorting Core at Baylor College of Medicine, which is funded from the CPRIT Core Facility Support Award (CPRIT-RP180672), the NIH (P30 CA125123 and S10 RR024574), and the expert assistance of Joel M Sederstrom for assistance with flow cytometry. Single cell library preparation, Illumina sequencing, and Cell Ranger alignment was facilitated by Advanced Technology Genomics Core at MD Anderson Cancer Research Center funded by CA016672(ATGC). IMARIS image analysis was performed using Rice University's Shared Equipment Authority (SEA) IMARIS workstation. We thank George Eisenhoffer and Oscar Ruiz (MD Anderson) for advice regarding flow cytometry and single-cell RNA-seq methodology. We thank Sarah Kucenas (University of Virginia) for helpful advice on glial populations. We thank Robert Naja and Robyn Fenty for technical assistance.

## Additional information

### Funding

| Funder | Grant reference number | Author |
| --- | --- | --- |
| Cancer Prevention and Research Institute of Texas | RR170062 | Rosa A Uribe |
| National Science Foundation | 1942019 | Rosa A Uribe |
| SDB | Choose Development! | Jessa L Westheimer |
| Houston Livestock Show and Rodeo Research Award | | Phillip A Baker<br>Joshua A Moore |

The funders had no role in study design, data collection and interpretation, or the decision to submit the work for publication.

### Author contributions

Aubrey GA Howard IV, Rodrigo Ibarra-García-Padilla, Joshua A Moore, Conceptualization, Data curation, Formal analysis, Validation, Investigation, Visualization, Methodology, Writing - original draft; Phillip A Baker, Conceptualization, Data curation, Formal analysis, Validation, Investigation, Visualization, Writing - original draft; Lucia J Rivas, Validation, Investigation, Visualization, Writing - original draft; James J Tallman, Data curation, Validation, Visualization, Writing - review and editing; Eileen W Singleton, Validation, Investigation, Methodology, Writing - review and editing; Jessa L Westheimer, Julia A Corteguera, Validation, Investigation, Visualization, Writing - review and editing; Rosa A Uribe, Conceptualization, Resources, Data curation, Formal analysis, Supervision, Funding acquisition, Validation, Investigation, Visualization, Methodology, Writing - original draft, Project administration, Writing - review and editing

### Author ORCIDs

Rosa A Uribe ⓘ https://orcid.org/0000-0002-0427-4493

### Ethics

Animal experimentation: All animal work was performed under protocols approved by, and in accordance with, the Rice University Institutional Animal Care and Use Committee (IACUC), under protocol 1144724.

Decision letter and Author response
Decision letter https://doi.org/10.7554/eLife.60005.sa1
Author response https://doi.org/10.7554/eLife.60005.sa2

## Additional files

### Supplementary files
• Transparent reporting form

### Data availability

Sequencing data have been added to GEO under accession GSE152906. The raw sequence read files and processed cellular barcode, gene, and matrix files produced by CellRanger are available in the National Center for Biotechnology Information's (NCBI) Gene Expression Omnibus (GEO) database (https://www.ncbi.nlm.nih.gov/geo/), accession number: GSE152906. The atlas/associated processed Seurat objects are available on the University of California, Santa Cruz (UCSC) Cell Browser (https://zebrafish-neural-crest-atlas.cells.ucsc.edu/). Code written for Seurat data analysis is available on GitHub (https://github.com/UribeLabRice/NeuralCrest_Atlas_2020; copy archived at https://archive.softwareheritage.org/swh:1:rev:195fe7dd0bf388ad4cd6f01db052f45d8b8fa67e/).

The following datasets were generated:

| Author(s) | Year | Dataset title | Dataset URL | Database and Identifier |
|---|---|---|---|---|
| Howard AA, Baker PA, Ibarra-García-Padilla R, Moore JA, Rivas LJ, Tallman JJ, Corteguera JA, Westheimer JL, Singleton EW, Uribe RA | 2020 | An atlas of neural crest lineages along the posterior developing zebrafish at single-cell resolution | https://www.ncbi.nlm.nih.gov/geo/query/acc.cgi?acc=GSE152906 | NCBI Gene Expression Omnibus, GSE152906 |
| Howard AA, Baker PA, Ibarra-García-Padilla R, Moore JA, Rivas LJ, Tallman JJ, Singleton EW, Westheimer JL, Corteguera JA, Uribe RA | 2020 | Zebrafish Neural Crest Atlas | https://zebrafish-neural-crest-atlas.cells.ucsc.edu/ | UC Santa Cruz Cell Browser, zebrafish-neural-crest-atlas |

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
