## [Decision Letter]

**Acceptance summary:**

The manuscript reports comprehensive analysis of single cell sequencing of FACS sorted neural crest cells from the posterior head and trunk of the zebrafish embryo at two developmental time points, generating a ~4,000 cell scRNAseq dataset. By performing standard analysis using Seurat to cluster and annotate the data, the authors identified major cell types within the neural crest at these stages, including pigment cells, craniofacial and enteric neuronal populations, and unique combinatorial *hox* gene expression among particular neural crest cell types. The results presented here could help other investigators identify populations of neural crest cells as well as how the gene regulatory network functions at later time points in development.

**Decision letter after peer review:**

Thank you for submitting your article "An atlas of neural crest lineages along the posterior developing zebrafish at single-cell resolution" for consideration by *eLife*. Your article has been reviewed by three peer reviewers, and the evaluation has been overseen by a Reviewing Editor and Richard White as the Senior Editor. The following individuals involved in review of your submission have agreed to reveal their identity: Cecilia Lanny Winata (Reviewer #2); Kristin Artinger (Reviewer #3).

The reviewers have discussed the reviews with one another and the Reviewing Editor has drafted this decision to help you prepare a revised submission.

Summary:

The manuscript reports comprehensive analysis of single cell sequencing of FACS sorted neural crest cells from the posterior head and trunk of the zebrafish embryo at two developmental time points, generating a ~4,000 cell scRNAseq dataset. By performing standard analysis using Seurat to cluster and annotate the data, the authors identified major cell types within the neural crest at these stages, including pigment cells, craniofacial and enteric neuronal populations, and unique combinatorial *hox* gene expression among particular neural crest cell types. The results presented here could help other investigators identify populations of neural crest cells as well as how the gene regulatory network functions at later time points in development.

Essential revisions:

There was consensus among the reviewers, that while this work presents a useful data set, the current manuscript falls short on conclusions about neural crest drawn from the data. The manuscript discusses at length about gene expression in various NCC populations without much functional analysis or offering new insights. To become suitable for publication in *eLife* the manuscript would need to be revised to shorten methodological aspects, present new findings and highlight their significance. Based on the data presented, enteric neurons and *hox* code present such opportunities. For enteric neurons, the current manuscript has missed the opportunity to use the scRNA-seq data in an unbiased way, the authors searched for known signatures. It will be important to demonstrate that this data set can reveal previously undescribed genes and pathways. It is understood that in the current situation of ongoing pandemic and reduced laboratory efforts, functional analyses are not feasible, although validation of expression of some novel genes, would significantly strengthen the manuscript.

1) The Introduction would be improved by a discussion about the known gene regulatory network at these stages and cell types. It will help put the study in context of what is known in the field. This would include expression analysis by RNA in situ and bulk RNA-seq analysis on this population.

2) The use of PTU does not seem to be necessary for the purpose of the experiment as the GFP signal should still be visible even with pigmentation. The reviewers therefore wondered why the authors incorporated this procedure. Although PTU is technically not supposed to affect the early steps of melanophore differentiation, possible implications of this need to be at least clarified as there is evidence that PTU can affect the organism at molecular and physiological levels.

3) There should be a discussion of how the cell type classification markers where chosen for the clustering. For those not in the field, it is not clear. Please add.

4) The Results section should be shortened and condensed significantly. Over half of the Results section (some 12 to 14 pages) is merely describing the authors' process of annotating the data. This could easily be summarized in a full-page figure with an annotated tSNE plot and a dot plot indicating the markers that were used to define the annotations. It would, in fact, be much clearer for readers.

5) Additionally, many other sections of the Results are so verbose that they obscure the authors' meaning. Tightening the prose would significantly improve the manuscript. Paragraph three of subsection “Pigment cell chromatophore lineages resolved” is such an example. Could the authors not just say "Previous work identified melanophores in distinct proliferative and differentiating states at 5 dpf (Saunders et al., 2019). We find these states arise between 50 and 68 hpf, as all melanophores were in the G1 phase at 48-50 hpf, but formed two distinct clusters at 68-70 hpf differentiated by their cell cycle state."

6) The HCR expression analyses are a nice addition to show the expression of some of the genes that are expressed in specific populations. However, the genes shown are already known and thus higher resolution images (For craniofacial figure, pigment I and J are hard to see, enteric figure is good but also would benefit from sectional analysis) to show neural crest cells at the cellular level and potential overall of expression would move the field and be more similar to the single cell sequencing. It would also be helpful for data interpretation. That being said, it would also be interesting to show some of the genes in the clusters that are novel or that have not been shown to be expressed in that cell type as well.

7) "Hierarchical clustering of cells with General Mesenchyme and Chondrogenic identities using a cluster tree": This is not technically correct, assuming the authors used BuildClusterTree (which it seems so from the Materials and methods). This function does not operate on the level of cells, but hierarchically clusters the mean expression signature of the already-calculated clusters.

8) It's not clear why the authors present the data twice, first as each individual time point and then as a combined analysis, when the same clusters are recovered in the combined analysis. It seems as if it would be easier and more informative to just present the combined data.

9) The authors provide an estimated number of cells reported by 10x Cell Ranger pipeline which corresponds to 2300 and 2580 for 48-50hpf and 68-70hpf, respectively. We understood that these numbers correspond to the estimated number of cells that 10x pipeline was able to identify which in the following steps serves as an input for Seurat analysis in R. The total number of usable cells after all QC steps were 1608 (58-60hpf) and 2410 (68-70hpf). However, there was no information on how many cells were loaded on to the 10x chip. This information would help others, especially in the zebrafish community, to better plan single-cell experiments and would provide additional information about cell suspension quality.

10) The reviewers were surprised that there is a large population of neural crest classified cells at these stages. They wondered if those are progenitor cells and do they remain in adults? In addition, it seems like most of the neuronal population in enteric, where are the dorsal root ganglia and sympathetic neuronal populations?

11) In the two stages of neural crest development, it would be interesting to include how the gene expression changes across developmental time. Could you present a combined UMAP with stages colored to see where stages fall and sets of genes that change expression through developmental time in subclusters? Highlighting genes that change over developmental time in each cluster would be very informative.

12) Discussion penultimate paragraph: This is potentially an interesting conclusion that the authors could spend more time on in the paper. Is it worth generating some version of a figure that combines a review of what is known about these signatures from other systems (and which other systems) with what the authors observe in zebrafish? One could envision some form of dot plot with an organismal color code, but obviously the authors may have more creative ideas.

13) (783/784) Mitochondrial genes were regressed out from the data set during the Cell Ranger alignment. Mitochondrial genes, together with ribosomal genes, are useful metrics for the assessment of cell quality. High expression levels of mitochondrial genes may indicate poor sample quality, apoptotic cells, or multiplets. Or if it is limited to a few clusters, it may reveal the nature of some cells (e.g. increased metabolic activity). However, the authors decided to remove mitochondrial genes from the alignment. Why?

14) The paper makes no mention of how the data and code will be made available. It would be standard practice to deposit the FASTQ files as well as the counts table output by CellRanger in GEO. Moreover, it's fairly standard practice (and generally appreciated) to make the processed Seurat objects available (either through lab website, Data Dryad, or some other hosting platform). Additionally, it's generally accepted practice to publish the code that was used to analyze the data and generate the figures.

[Editors' note: further revisions were suggested prior to acceptance, as described below.]

Thank you for submitting your article "An atlas of neural crest lineages along the posterior developing zebrafish at single-cell resolution" for consideration by *eLife*. Your article has been reviewed by three peer reviewers, and the evaluation has been overseen by a Reviewing Editor and Richard White as the Senior Editor. The following individuals involved in review of your submission have agreed to reveal their identity: Cecilia Lanny Winata (Reviewer #2); Kristin Artinger (Reviewer #3).

The reviewers have discussed the reviews with one another and the Reviewing Editor has drafted this decision to help you prepare a revised submission.

Summary:

The manuscript reports comprehensive analysis of single cell sequencing of FACS sorted neural crest cells from the posterior head and trunk of the zebrafish embryo at two developmental time points, generating a ~4,000 cell scRNAseq dataset. By performing standard analysis using Seurat to cluster and annotate the data, the authors identified major cell types within the neural crest at these stages, including pigment cells, craniofacial and enteric neuronal populations, and unique combinatorial *hox* gene expression among particular neural crest cell types. The results presented here could help other investigators identify populations of neural crest cells as well as how the gene regulatory network functions at later time points in development.

Revisions:

All reviewers thought that the authors have made significant improvement to the manuscript by removing many of the technical descriptions from their results. However they also thought that more can be done to make the Results section more readable. Many unnecessary details and repetitions remain, which obscures the communication of essential findings. Moreover, the manuscript is perceived as written for a neural crest audience and thus less accessible to a more general (*eLife*) audience making the authors motivation and essential findings obscured. The specific comments and suggestions of the reviewers that can guide additional revisions are appended below.

Reviewer #1:

My only remaining comment that should not block publication, but perhaps would inspire some additional revisions from the authors, is that overall, I still find that the prose is disappointingly focused on technical aspects of a pretty standard single-cell analysis at the expense of describing discoveries from those analyses. Many sections of the paper end with the main conclusion that the section "demonstrates the power of the sox10 atlas to.…" but I don't know why this is repeatedly considered a result, given that standard and well demonstrated analysis methods were used on data generated using now standard approaches. The new and expanded Discussion section illustrates that the authors have found some interesting results, but I feel that they dilute them significantly by focusing so much on telling us about the power of now-standard single-cell genomics approaches and about the process of annotating the data, when they could instead place the focus on their more interesting findings. For instance, I find this is especially true in the presentation of the mesenchyme - which seems mostly focused on describing existing mesenchyme markers and how they were used to identify the mesenchyme and then the number of clusters found therein, but without significant description of what those clusters are or what defines them.

Reviewer #2:

Despite the new additions, the manuscript remains highly descriptive and a major issue in terms of writing style remains.

1) The authors have made significant improvement by removing many of the technical descriptions from their results. However this did little to make the Results section more readable. Many unnecessary details and repetitions remain, which obscures the communication of essential findings. The language used is often colloquial, with some grammatical errors which makes reading extremely challenging, especially for a non-expert in NCC or single cell biology. Having the manuscript read by a professional scientific editing service might be of help. To cite some examples:

– "We have utilized the Tg(-4.9sox10:EGFP) (hereafter referred to as 117 sox10:GFP) transgenic fish line to identify". Authors were requested to use the term "zebrafish" instead of "fish" since other fishes are used for these types of studies and it is confusing. Please correct this.

– What does "remarkably captured" mean?

– "Taken together, these results show that the scRNA-seq datasets effectively identify discrete subpopulations, and coupled with our HCR analysis, effectively shows we are able to validate these cell populations in vivo". Please clarify which subpopulations are referred to here.

– It is unclear what the authors meant by "early neuronal differentiation". Please specify the exact or range of developmental stage(s) for clarity.

– "Further investigation into these pathways led to the identification of…" – would it be sufficient to simply state for e.g. that "oprl1 and oprd1b, which are members of this pathway, were expressed in subcluster 3"?

– ". patterns of Homeobox transcription factors, known as *hox* genes…" – how can transcription factors be known as genes? Please rephrase this.

2) The Discussion section is also extremely lengthy, containing many repetitions of the results which could be significantly condensed. There are also a lot of anecdotal information written on opioids – which seems irrelevant and unnecessary.

3) In describing the mesenchymal clusters, the authors used the term "subtypes". It seems more appropriate to call them simply as "clusters" as there were no systematic analyses done (e.g. expression of specific cell type markers) to define whether each of these barx1+ and dlx2a+ clusters represent distinct cell subtypes. Moreover, the number of clusters obtained depends on the threshold set in the scRNA-seq data analysis, which is therefore arbitrary in nature.

Along the same lines, the resolution parameter from Seurat FindClusters function determines the number of obtained clusters. Naturally, the selection of this parameter is arbitrary and needs to be individually tailored for each sample. Looking at the data at different resolutions may help to improve the identification of novel cell types. Is there any particular reason why the authors decided to apply the resolution of 1.2? A clarification on how this threshold was chosen would be helpful.

4) "… pbx3b expression may promote the assumption of an IPAN signature characterized by the presence of calb2a, ache, and slc18a3a and the loss of inhibitor markers nos1 and vipb." – it is unclear how the authors' observations led to this hypothesis as earlier on they stated that "…pbx3b expression was found in combination with calb2a, vipb, and nos1…". Please clarify.

5) "Overall, these results suggest that sub-cluster 0 cells may represent a pool of immature sympatho-enteric neurons, and that sub-clusters 1 and 4 both represent better resolved, yet still immature, pools of enteric and sympathetic neurons." – it is unclear how the authors arrived at this notion, especially since cluster 0, 1, and 4 was not even mentioned at all previously.

6) It would be helpful if the authors also annotate the clusters in Figure 6—figure supplement 2 panel A and Figure 7—figure supplement 1 panel J.

7) "We performed cell filtering and clustering (.) cells which contained low (<200) or high (>2500) genes were removed from analysis".

– As indicated in Figure 1—figure supplement 1 panel C, around 4905 and 4669 cells were loaded for encapsulation into 10x cartridge, resulting in 2300 and 2580 identified cells. After a filtering step that was done on the basis of gene content only, 1608 and 2410 cells were obtained. The expected multiplet rate at such cell suspension concentration should be low (around 2% according to 10x protocol), therefore the great loss of cell numbers is rather surprising. Have the authors checked why the cell loss at the applied threshold level for the 48-50 hpf sample is much higher than expected. Also, did they check the distribution of expressed genes before setting up the thresholds for gene content per cell?

– Additionally, including mitochondrial gene content into the filtering step may help to confidently identify low quality cells.

8) Figure 1—figure supplement 1 panel C – inconsistent number format (comma, dot, space), e.g. mean read per cell (74,017 vs 62109 etc.)

9) “…Major cell type categories were based on the presence of signature marker genes…”. Is the word "presence" an appropriate term in this context? Presence is rather a binary value and takes two options: 1- present, 0 – not present. It seems more accurate to use the term "expression" of these signatures/marker genes.

---

## [Author Response]

Essential revisions:There was consensus among the reviewers, that while this work presents a useful data set, the current manuscript falls short on conclusions about neural crest drawn from the data. The manuscript discusses at length about gene expression in various NCC populations without much functional analysis or offering new insights. To become suitable for publication in eLife the manuscript would need to be revised to shorten methodological aspects, present new findings and highlight their significance. Based on the data presented, enteric neurons and hox code present such opportunities. For enteric neurons, the current manuscript has missed the opportunity to use the scRNA-seq data in an unbiased way, the authors searched for known signatures. It will be important to demonstrate that this data set can reveal previously undescribed genes and pathways. It is understood that in the current situation of ongoing pandemic and reduced laboratory efforts, functional analyses are not feasible, although validation of expression of some novel genes, would significantly strengthen the manuscript.

We have revised the manuscript to balance methodological aspects, while highlighting new findings made possible by leveraging analyses of our single cell RNA-seq datasets. We agree with reviewers that enteric and Hox gene expression patterns are opportunities to focus on novel findings; we have thus expanded analyses of these signatures in the revised manuscript.

First, we present analysis of incipient enteric neuron populations, where we have discovered the presence of cells along a spectrum of differentiation states by using sub-clustering analysis. From this analysis (Figure 5), we found that enteric populations in zebrafish exist in several sub-populations—enteric neuronal progenitors, immature enteric neurons and enteric cells differentiating towards a specific sensory subtype. Leveraging data from the enteric subpopulations we discovered, we performed unbiased pathway analysis and, among many new pathways, detected the presence of opioid pathway receptors within differentiating enteric neurons—a novel finding for the developmental ENS field. Importantly, we validated in situ expression of the opioid receptor genes within enteric neurons along the gut in vivo using HCR. We highlight and discuss the significance of opioid receptor detection within nascent enteric neurons within our revised discussion.

Secondly, we expanded analysis of Hox gene codes within the *sox10* atlas presented in the manuscript. By calculating the pairwise, co-expression values of all *hox* genes expressed within the atlas, we present a data matrix that summarizes combinatorial *hox* expression within enteric neurons and autonomic neurons (Figure 7). We found for the first time in zebrafish a specific enteric neuron *hox* code, marked by enrichment of *hoxb2a*, *hoxa5a*, *hoxd4a*, *hoxb5a* and *hoxb5b.* We validated in situ HCR expression of four of these *hox* factors within *phox2bb*+ enteric cells along the zebrafish midgut in vivo. The pairwise co-expression analysis indicated that enteric neurons present with varying combinatorial states for the enteric code during their differentiation, revealing novel details regarding heterogeneity of early enteric transcriptional states.

Finally, we have demonstrated that the *sox10* atlas can be used to showcase novel genes within subpopulations across time. We highlighted known and novel genes for the mesenchymal, pigment and neural populations in Figure 6—figure supplement 2. Extending beyond this and bringing the power of the atlas to the scientific community, we have deposited our merged Seurat atlas dataset to the UCSC Cell Browser, which enables interactive assessment of annotated cell populations and interrogation of genes of interest.

1) The Introduction would be improved by a discussion about the known gene regulatory network at these stages and cell types. It will help put the study in context of what is known in the field. This would include expression analysis by RNA in situ and bulk RNA-seq analysis on this population.

We have added in some examples of recent studies addressing gene regulatory networks of neural crest cells and their early forming derivatives. As highlighted in the Introduction, most of what is known regarding neural crest development using transcriptomics and chromatin profiling studies centers largely on early neural crest specification in pre-migratory and early migratory stages. Filling a gap in knowledge in the field, our study focuses on later phases of neural crest migration and lineage subtype differentiation.

2) The use of PTU does not seem to be necessary for the purpose of the experiment as the GFP signal should still be visible even with pigmentation. The reviewers therefore wondered why the authors incorporated this procedure. Although PTU is technically not supposed to affect the early steps of melanophore differentiation, possible implications of this need to be at least clarified as there is evidence that PTU can affect the organism at molecular and physiological levels.

We have added a sentence in the Materials and methods section to clarify the use of PTU in our studies. We added low dose PTU to embryos after the first day in development to prevent pigmentation formation that enabled ease of GFP sorting prior to dissections. While high dose PTU prior to the first day in development has been shown to adversely affect embryonic development, we have acknowledged this in the Materials and methods and that low dose PTU is not believed to cause defects in neural crest development at the stages we examined in our study.

3) There should be a discussion of how the cell type classification markers where chosen for the clustering. For those not in the field, it is not clear. Please add.

We have added information regarding how clusters were annotated with cell type classifications into the Results and the Materials and methods. We would also like to clarify that Seurat clustering of the single cell RNA-seq transcriptomes was performed based on PC analysis prior to any cell type classifications, standard for the Seurat pipeline, as we describe in the Materials and methods.

4) The Results section should be shortened and condensed significantly. Over half of the Results section (some 12 to 14 pages) is merely describing the authors' process of annotating the data. This could easily be summarized in a full-page figure with an annotated tSNE plot and a dot plot indicating the markers that were used to define the annotations. It would, in fact, be much clearer for readers.

We agree and have shortened and condensed the first half of the Results section. Along with that, we have also summarized/condensed the first original three figures into one major figure with one supplement. New Figure 1 contains annotated tSNE and dot plots summarizing major cell type categories of *sox10*+ cells for each major time point, as suggested by reviewers.

5) Additionally, many other sections of the results are so verbose that they obscure the authors' meaning. Tightening the prose would significantly improve the manuscript. Paragraph three of subsection “Pigment cell chromatophore lineages resolved” is such an example. Could the authors not just say "Previous work identified melanophores in distinct proliferative and differentiating states at 5 dpf (Saunders et al., 2019). We find these states arise between 50 and 68 hpf, as all melanophores were in the G1 phase at 48-50 hpf, but formed two distinct clusters at 68-70 hpf differentiated by their cell cycle state."

Along with comment 4 above, we have also edited many sections of the results to tighten prose, including the section regarding pigment cells.

6) The HCR expression analyses are a nice addition to show the expression of some of the genes that are expressed in specific populations. However, the genes shown are already known and thus higher resolution images (For craniofacial figure, pigment I and J are hard to see, enteric figure is good but also would benefit from sectional analysis) to show neural crest cells at the cellular level and potential overall of expression would move the field and be more similar to the single cell sequencing. It would also be helpful for data interpretation. That being said, it would also be interesting to show some of the genes in the clusters that are novel or that have not been shown to be expressed in that cell type as well.

We have replaced the HCR images for the mesenchyme craniofacial figure (new figure 3) with cropped (further zoomed in) images to illustrate the distinct expression domains of the mesenchymal markers we wish to highlight. We have replaced the HCR images showing pigment markers in I and J with new images (new Figure 2). We have added new HCR image data to show co-expression of novel genes of interest in the developing enteric neuronal population along the zebrafish gut, including genes of the Opioid family and Hox transcription factors. We thank reviewers for these suggestions.

7) "Hierarchical clustering of cells with General Mesenchyme and Chondrogenic identities using a cluster tree": This is not technically correct, assuming the authors used BuildClusterTree (which it seems so from the Materials and methods). This function does not operate on the level of cells, but hierarchically clusters the mean expression signature of the already-calculated clusters.

Thank you for pointing out this error. We indeed used the BuildClusterTree tool in Seurat to generate our cluster trees in the paper. We have clarified description of this in the text.

8) It's not clear why the authors present the data twice, first as each individual time point and then as a combined analysis, when the same clusters are recovered in the combined analysis. It seems as if it would be easier and more informative to just present the combined data.

We endeavored to show each dataset separately first, as we intended to showcase the *sox10* cellular populations present at each time point to then allow for an appreciation of the combined dataset later shown. Indeed, we pulled important data from individual time point data, as evidenced by the analysis of enteric sub-clusters and the populations contained within them at the late time point (Figure 5). On the other hand, we leveraged the combined analysis to ask questions about global *hox* gene expression trends (Figure 7) and differential expression between time points (Figure 6). This demonstrates that examination of the dataset at multiple scales yields interesting results worth examining.

9) The authors provide an estimated number of cells reported by 10x Cell Ranger pipeline which corresponds to 2300 and 2580 for 48-50hpf and 68-70hpf, respectively. We understood that these numbers correspond to the estimated number of cells that 10x pipeline was able to identify which in the following steps serves as an input for Seurat analysis in R. The total number of usable cells after all QC steps were 1608 (58-60hpf) and 2410 (68-70hpf). However, there was no information on how many cells were loaded on to the 10x chip. This information would help others, especially in the zebrafish community, to better plan single-cell experiments and would provide additional information about cell suspension quality.

We thank the reviewers for this suggestion. We have added this information into the Figure 1—figure supplement 1 and in the Materials and methods.

10) The reviewers were surprised that there is a large population of neural crest classified cells at these stages. They wondered if those are progenitor cells and do they remain in adults? In addition, it seems like most of the neuronal population in enteric, where are the dorsal root ganglia and sympathetic neuronal populations?

Yes, we had wondered the same after initial examination of the datasets. We do not think the cells we classified as neural crest are merely progenitor cells. They express bona fide neural crest markers such as *sox10*, *foxd3, ngfrb* and *tfap2a*, and a zebrafish-specific neural crest marker, *crestin*. We confirmed *crestin* co-expression with *foxd3* and *ngfrb* via HCR (Figure 4) and have also seen *crestin*+ cells in neural crest pathways at these time points before (Uribe et al., 2015, et al., 2018). We believe they are bona fide neural crest based on their signatures. We don’t know if neural crest cells with these signatures are remaining in adult zebrafish, it would certainly be interesting to explore in the future.

In terms of detecting dorsal root ganglia and sympathetic neuronal cells within the *sox10* datasets, we have indeed detected dorsal root ganglia progenitor signatures in our first time point, which we described as sensory signatures in the results and shown in Figure 1—figure supplement 6. For sympathetic neuronal populations, upon closer examination of the second time point of our datasets, we have detected sympathetic signatures and created a new figure to describe this, Figure 5—figure supplement 2. We thank reviewers for the suggestions.

11) In the two stages of neural crest development, it would be interesting to include how the gene expression changes across developmental time. Could you present a combined UMAP with stages colored to see where stages fall and sets of genes that change expression through developmental time in subclusters? Highlighting genes that change over developmental time in each cluster would be very informative.

We agree with reviewers. We had originally included a combined UMAP with stages colored to see where stages fall within the atlas, in original Figure 8—figure supplement 1A. We now have expanded this analysis in a new figure, Figure 6—figure supplement 2, to include sets of genes that change expression in sub-clusters over time, as suggested by the reviewers. For this, we focused on subsets of the atlas to examine mesenchyme, pigment and neural/neuronal annotated cellular populations. From this we found expected genes from our prior analysis, as well as novel genes that have not been described before in these *sox10*+ cell populations and highlight them in the results. Globally, we agree with reviewers that being able to leverage the atlas to examine changing gene expression over time is informative, and thus, we have integrated the dataset into an interactive cell browser, hosted at the UC Santa Cruz Cell browser (described in the paper) where users can interactively view global cell populations, as well as genes of interest. We thank reviewers and hope this resource will help move the field forward.

12) Discussion penultimate paragraph: This is potentially an interesting conclusion that the authors could spend more time on in the paper. Is it worth generating some version of a figure that combines a review of what is known about these signatures from other systems (and which other systems) with what the authors observe in zebrafish? One could envision some form of dot plot with an organismal color code, but obviously the authors may have more creative ideas.

We have edited this statement out of our writing in the Discussion appropriately. While we agree that this is an interesting figure to synthesize on the topic, it is beyond the scope of what we aimed for our atlas and we intend to include this great suggestion in a future review article or future paper.

13) Mitochondrial genes were regressed out from the data set during the Cell Ranger alignment. Mitochondrial genes, together with ribosomal genes, are useful metrics for the assessment of cell quality. High expression levels of mitochondrial genes may indicate poor sample quality, apoptotic cells, or multiplets. Or if it is limited to a few clusters, it may reveal the nature of some cells (e.g. increased metabolic activity). However, the authors decided to remove mitochondrial genes from the alignment. Why?

We thank reviewers for catching this error in our methods description. We clarify that we did not remove mitochondrial genes from the alignment and have removed this statement from the Materials and methods.

14) The paper makes no mention of how the data and code will be made available. It would be standard practice to deposit the FASTQ files as well as the counts table output by CellRanger in GEO. Moreover, it's fairly standard practice (and generally appreciated) to make the processed Seurat objects available (either through lab website, Data Dryad, or some other hosting platform). Additionally, it's generally accepted practice to publish the code that was used to analyze the data and generate the figures.

We thank the reviewers for catching this omission. Indeed, we have deposited the FASTQ datasets to GEO, the details of which are now included at the end of the Materials and methods. Further, we have transformed the Seurat datasets into an interactive cell browser, hosted at the UC Santa Cruz Cell browser (described in the paper) where users can interactively view global cell populations, as well as genes of interest. The code used to analyze datasets are also available on our lab GitHub. We thank reviewers and hope this resource will help move the field forward.

[Editors' note: further revisions were suggested prior to acceptance, as described below.]

Revisions:All reviewers thought that the authors have made significant improvement to the manuscript by removing many of the technical descriptions from their results. However they also thought that more can be done to make the Results section more readable. Many unnecessary details and repetitions remain, which obscures the communication of essential findings. Moreover, the manuscript is perceived as written for a neural crest audience and thus less accessible to a more general (eLife) audience making the authors motivation and essential findings obscured. The specific comments and suggestions of the reviewers that can guide additional revisions are appended below.Reviewer #1:My only remaining comment that should not block publication, but perhaps would inspire some additional revisions from the authors, is that overall, I still find that the prose is disappointingly focused on technical aspects of a pretty standard single-cell analysis at the expense of describing discoveries from those analyses. Many sections of the paper end with the main conclusion that the section "demonstrates the power of the sox10 atlas to.…" but I don't know why this is repeatedly considered a result, given that standard and well demonstrated analysis methods were used on data generated using now standard approaches. The new and expanded Discussion section illustrates that the authors have found some interesting results, but I feel that they dilute them significantly by focusing so much on telling us about the power of now-standard single-cell genomics approaches and about the process of annotating the data, when they could instead place the focus on their more interesting findings. For instance, I find this is especially true in the presentation of the mesenchyme - which seems mostly focused on describing existing mesenchyme markers and how they were used to identify the mesenchyme and then the number of clusters found therein, but without significant description of what those clusters are or what defines them.

We have edited the presentation of the mesenchyme results to focus on what we discovered defines the clusters, driving the annotations. Following annotations, we found various specific gene signatures among the mesenchyme clusters. For example, we found mesenchyme-identity cells expressing chondrogenic markers (barx1), proliferative signatures (pcna), migratory genes (snai2, rac1), stem-like signatures (uhrf1), etc. Description of what the clusters are and what genes defines them are included in the annotation table in Figure 1—figure supplement 5. For clarity, we now refer to the various mesenchyme clusters as transcriptionally-distinct mesenchyme subpopulations and/or clusters and have edited the text to reflect this. We leave further investigation of the mesenchyme populations open to researchers interested in the future.

We thank the reviewer for providing feedback to allow us to further improve the text. Overall, we have made edits throughout and removed a majority of the language that centered upon highlighting the technical methods of the analysis. We have edited the Results and Discussion sections of the manuscript to be more concise, while also ensuring sufficient details were maintained for the reader to understand the datasets. We have re-organized the discussion to highlight novel findings.

Reviewer #2:Despite the new additions, the manuscript remains highly descriptive and a major issue in terms of writing style remains.1) The authors have made significant improvement by removing many of the technical descriptions from their results. However this did little to make the Results section more readable. Many unnecessary details and repetitions remain, which obscures the communication of essential findings. The language used is often colloquial, with some grammatical errors which makes reading extremely challenging, especially for a non-expert in NCC or single cell biology.

We thank the reviewer for providing feedback to allow us to further improve the text. Overall, we have made edits throughout and removed a majority of the language that centered upon highlighting the technical methods of the analysis. We have edited the Results and Discussion sections of the manuscript to be more concise, while also ensuring sufficient details were maintained for the reader to understand the datasets. We have re-organized the discussion to highlight novel findings.

Having the manuscript read by a professional scientific editing service might be of help. To cite some examples:– "We have utilized the Tg(-4.9sox10:EGFP) (hereafter referred to as 117 sox10:GFP) transgenic fish line to identify". Authors were requested to use the term "zebrafish" instead of "fish" since other fishes are used for these types of studies and it is confusing. Please correct this.

We removed the word “fish line” and replaced it with “transgenic line” in this sentence.

– What does "remarkably captured" mean?

We removed the word “remarkably”.

– "Taken together, these results show that the scRNA-seq datasets effectively identify discrete subpopulations, and coupled with our HCR analysis, effectively shows we are able to validate these cell populations in vivo". Please clarify which subpopulations are referred to here.

We have edited this conclusion statement to clarify which populations (pigment cells) are referenced.

– It is unclear what the authors meant by "early neuronal differentiation". Please specify the exact or range of developmental stage(s) for clarity.

We have clarified the range of developmental stages early neuronal differentiation refers to with regard to enteric cells discussed, which is now included in a revised paragraph.

– "Further investigation into these pathways led to the identification of…" – would it be sufficient to simply state for e.g. that "oprl1 and oprd1b, which are members of this pathway, were expressed in subcluster 3"?

We have edited the paragraph describing opioid pathway gene expression within enteric populations for clarity throughout.

– ". patterns of Homeobox transcription factors, known as hox genes…" – how can transcription factors be known as genes? Please rephrase this.

We have edited this phrasing to state, “expression of *hox* genes, which encode for Homeobox transcription factors…”.

2) The Discussion section is also extremely lengthy, containing many repetitions of the results which could be significantly condensed. There are also a lot of anecdotal information written on opioids which seems irrelevant and unnecessary.

We have performed edits to make the Discussion section more concise than the previous manuscript version. We have removed anecdotal information regarding opioids. Overall, we have made edits throughout the Discussion and removed all the repetitions we could find.

3) In describing the mesenchymal clusters, the authors used the term "subtypes". It seems more appropriate to call them simply as "clusters" as there were no systematic analyses done (e.g. expression of specific cell type markers) to define whether each of these barx1+ and dlx2a+ clusters represent distinct cell subtypes. Moreover, the number of clusters obtained depends on the threshold set in the scRNA-seq data analysis, which is therefore arbitrary in nature.

The reviewer has pointed an area worth further clarification. We originally called the cells “subtypes” due to the fact that they comprised the clusters we had designated with the major cell type identity of mesenchyme, yet also exhibited transcriptionally-distinct signatures which we carefully annotated by various marker genes denoting cell states. The table in Figure 1—figure supplement 5 lists the major cell type designation for each cluster, as well as individual annotations for each cell “subtype”, under each major cell type category. Following all annotations, we found various specific gene signatures among the mesenchyme clusters. For example, we found mesenchyme-identity cells expressing chondrogenic markers, proliferative signatures, migratory genes, stem-like signatures, etc. For clarity, we will refer to the various mesenchyme clusters as transcriptionally-distinct mesenchyme subpopulations and/or clusters and have edited the text to reflect this.

Along the same lines, the resolution parameter from Seurat FindClusters function determines the number of obtained clusters. Naturally, the selection of this parameter is arbitrary and needs to be individually tailored for each sample. Looking at the data at different resolutions may help to improve the identification of novel cell types. Is there any particular reason why the authors decided to apply the resolution of 1.2? A clarification on how this threshold was chosen would be helpful.

As one of the first data analysis methods we enacted when examining our datasets, we indeed looked at the data at different resolutions to help improve identification of novel cell types. We have added a sentence in the Materials and methods to describe this, ”The resolution value was empirically determined for the FindClusters by iterative assessment of cluster patterns generated values ranging from 0.4 to 2.0. The resolution was set to 1.2 which balanced between over segmentation of cell identities and effective separation of tSNE-mapped clusters.”

4) "… pbx3b expression may promote the assumption of an IPAN signature characterized by the presence of calb2a, ache, and slc18a3a and the loss of inhibitor markers nos1 and vipb." – it is unclear how the authors' observations led to this hypothesis as earlier on they stated that "…pbx3b expression was found in combination with calb2a, vipb, and nos1…". Please clarify.

We have edited the paragraph in the Results describing the IPAN signature expression in zebrafish to address this concern and clarify the data. Furthermore, we moved description of the model of IPAN fate acquisition (Figure 5F) to the Discussion, where it is more appropriately described in conjunction with the recent literature in mouse.

5) "Overall, these results suggest that sub-cluster 0 cells may represent a pool of immature sympatho-enteric neurons, and that sub-clusters 1 and 4 both represent better resolved, yet still immature, pools of enteric and sympathetic neurons." – it is unclear how the authors arrived at this notion, especially since cluster 0, 1, and 4 was not even mentioned at all previously.

We agree inclusion of more information about sub-clusters 0, 1 and 4 would strengthen and clarify the sub-cluster results. We have expanded this paragraph in the Results section to describe sub-clusters 0, 1 and 4 in more detail.

6) It would be helpful if the authors also annotate the clusters in Figure 6—figure supplement 2 panel A and Figure 7—figure supplement 1 panel J.

We agree. We have annotated the clusters in Figure 6—figure supplement 2A and Figure 7—figure supplement 1J.

7) "We performed cell filtering and clustering (.) cells which contained low (<200) or high (>2500) genes were removed from analysis".– As indicated in Figure 1—figure supplement 1 panel C, around 4905 and 4669 cells were loaded for encapsulation into 10x cartridge, resulting in 2300 and 2580 identified cells. After a filtering step that was done on the basis of gene content only, 1608 and 2410 cells were obtained. The expected multiplet rate at such cell suspension concentration should be low (around 2% according to 10x protocol), therefore the great loss of cell numbers is rather surprising. Have the authors checked why the cell loss at the applied threshold level for the 48-50 hpf sample is much higher than expected. Also, did they check the distribution of expressed genes before setting up the thresholds for gene content per cell?

We investigated the distribution of expressed genes prior to setting a final threshold for our analysis, considering appropriate treatment for both time points analyzed. As we would have needed to apply the <2500 gene threshold to the 68-70 hpf dataset, we also accepted this value at the 48-50 hpf dataset to ensure as equal treatment between the two as possible. Selecting a relatively stringent threshold for the number of features (<200 and >2500) allowed us to produce a high-quality dataset. While it is possible that this diminishes discovering possible rare cell populations, the stringent datasets enabled us to confidently investigate the transcriptional landscape of neural crest. Re-evaluation of our 48-50 hpf dataset with a higher threshold (>200 and < 4000) retains a higher fraction of cells (2297 cells), but the inclusion of these cells would not significantly alter our subsequent analysis. At this point we feel that erring on the more stringent side to produce a high-quality dataset is preferrable to the reciprocal, especially when we ultimately intend for our analysis to serve as a basis for the community to use moving forward. Should information contained within those cells be of useful to a user in the future, they have been included in the raw data published on the NCBI’s Gene Expression Omnibus portal.

– Additionally, including mitochondrial gene content into the filtering step may help to confidently identify low quality cells.

We are grateful for the suggestion about using mitochondrial gene content in the filtering step to assist in detecting low quality cells. We have determined that the removal of the cells which express moderate levels of mitochondrial genes minimally impacts the dataset, particularly when paired with the strict threshold described above. To this effect, we have retained cells with appreciable levels of mitochondrial gene expression and feel confident that the resulting dataset is robustly reflective of the diverse transcriptional landscape expected in posterior neural crest at the stages sampled.

8) Figure 1—figure supplement 1 panel C – inconsistent number format (comma, dot, space), e.g. mean read per cell (74,017 vs 62109 etc.)

We have removed all commas in the table within Figure 1—figure supplement 1 to utilize consistent number formatting.

9) “…Major cell type categories were based on the presence of signature marker genes…”. Is the word "presence" an appropriate term in this context? Presence is rather a binary value and takes two options: 1- present, 0 – not present. It seems more accurate to use the term "expression" of these signatures/marker genes.

We have changed the word “presence” and instead use “expression”.